# Notch-mediated conversion of activated T cells into stem cell memory-like T cells for adoptive immunotherapy

Taisuke Kondo[1], Rimpei Morita[1], Yuumi Okuzono[2], Hiroko Nakatsukasa[1], Takashi Sekiya[1], Shunsuke Chikuma[1], Takashi Shichita[1], Mitsuhiro Kanamori[1], Masato Kubo[3,4], Keiko Koga[2], Takahiro Miyazaki[2], Yoshiaki Kassai[2] & Akihiko Yoshimura[1]

Adoptive T-cell immunotherapy is a promising approach to cancer therapy. Stem cell memory T ($T_{SCM}$) cells have been proposed as a class of long-lived and highly proliferative memory T cells. CD8$^+$ $T_{SCM}$ cells can be generated *in vitro* from naive CD8$^+$ T cells via Wnt signalling; however, methods do not yet exist for inducing $T_{SCM}$ cells from activated or memory T cells. Here, we show a strategy for generating $T_{SCM}$-like cells *in vitro* (i$T_{SCM}$ cells) from activated CD4$^+$ and CD8$^+$ T cells in mice and humans by coculturing with stromal cells that express a Notch ligand. i$T_{SCM}$ cells lose PD-1 and CTLA-4 expression, and produce a large number of tumour-specific effector cells after restimulation. This method could therefore be used to generate antigen-specific effector T cells for adoptive immunotherapy.

[1] Department of Microbiology and Immunology, Keio University School of Medicine, 35 Shinanomachi, Shinjuku-ku, Tokyo 160-8582, Japan. [2] Inflammation Drug Discovery Unit, Pharmaceutical Research Division, Takeda Pharmaceutical Company Limited, 26-1 Muraoka-Higashi 2-chome, Fujisawa-shi, Kanagawa 251-8555, Japan. [3] Division of Molecular Pathology, Research Institute for Biomedical Science, Tokyo University of Science, 2641 Yamazaki, Noda-shi, Chiba 278-8510, Japan. [4] Laboratory for Cytokine Regulation, RIKEN Center for Integrative Medical Sciences (IMS), RIKEN Yokohama Institute, 1-7-22 Suehiro-cho, Tsurumi-ku, Yokohama City, Kanagawa 230-0045, Japan. Correspondence and requests for materials should be addressed to A.Y. (email: yoshimura@a6.keio.jp).

Adoptive T-cell transfer to treat cancer has been applied in clinical settings, but the effects have been limited because activated T cells are short-lived and easily lose their functionality[1,2]. Indeed, acquired tumour-specific T cells are often already exhausted in tumour microenvironments, and the enhancement of effector function *in vitro* paradoxically attenuates the *in vivo* antitumour efficacy of transferred T cells[3,4]. To overcome these obstacles, rejuvenated functional cytotoxic T lymphocytes (CTLs) can be generated from induced pluripotent stem (iPS) cells derived from antigen-specific T cells[5,6], a strategy that provides an unlimited supply of antigen-specific CTLs. However, T cell receptor (TCR) repertoires are often restricted in the process of iPS cell generation, which may result in the failure to react to a broad variety of tumour-associated antigens (TAAs). Transplantation of multiple TAA-reactive and expandable T cells may be efficacious against antigenic-drifted tumour cells that evade destruction by CTLs.

T cell populations have been classified by several surface markers and distinguished by their functions and residency, along with their effector cytokine production. Effector memory T ($T_{EM}$) cells and central memory T ($T_{CM}$) cells circulate in the blood and target the secondary lymphoid tissues[7]. Resident memory T ($T_{RM}$) cells remain at local sites to respond immediately to secondary infection. These cells can rapidly produce multiple functional molecules after restimulation to control the invasion and spread of pathogens. However, $T_{EM}$ and $T_{RM}$ cells have a limited potential for population expansion, and tend to become terminally differentiated and subsequently exhausted. Naive T cells, which have not been exposed to antigens, resist terminal differentiation and exhaustion when compared with memory T cells, and maintain strong proliferative potential after antigen stimulation. Therefore, compared with other subsets, naive T cells may be superior for adoptive immunotherapy[8]. However, the number of naive T cells that are specific for any given antigen is very low compared with memory T cells. Thus, antigen-specific memory T cells that have naive-like phenotypes are ideal for adoptive T cell therapy.

T cell subsets were originally classified into naive, effector and several memory cell populations[9]. However, the classification does not fully represent the contribution of T cell populations in infectious diseases, cancer, and disorders of ageing, and technical advances have revealed the existence of previously unknown T cell subsets. For example, memory stem cells ($T_{SCM}$) and memory cells with naive phenotypes ($T_{MNP}$) have been detected in T cell populations previously characterized as being naive[10,11]. Both $T_{SCM}$ and $T_{MNP}$ cells, which express naive T cell markers ($CD45RA^+CD45RO^-CCR7^+CD62L^+$), respond rapidly to antigens, express multiple effector molecules and produce memory and effector cells. Notably, these subsets have greater proliferative potential than naive cells.

The method by which $T_{SCM}$ cells are generated both *in vivo* and *in vitro* is not well established. Gattinoni *et al.* found that mouse and human $CD8^+$ $T_{SCM}$ cells can be generated effectively *in vitro* from naive $CD8^+$ T cells by stimulating the TCR in the presence of Wnt3A or inhibitors of glycogen synthase kinase-3β (GSK-3β)[10,12]. We confirmed that a GSK-3β inhibitor generated $CD8^+$ $T_{SCM}$ cells from naive T cells *in vitro*, yet this method failed to generate $CD4^+$ and $CD8^+$ $T_{SCM}$ cells from activated T cells. While verifying the effects of signalling molecules on T cells, we discovered that Notch signalling converts activated T cells into $T_{SCM}$-like cells (called '$iT_{SCM}$' cells) in mice and humans. These $iT_{SCM}$ cells are generated by coculturing of activated T cells with Notch ligand-expressing stromal cells. $iT_{SCM}$ cells respond to antigen restimulation with greater expanding potential than other T cell subsets. $iT_{SCM}$ cells also have a long-lived and self-renewing potential, are resistant to cell cycle arrest and apoptosis after TCR stimulation, and demonstrate great antitumour activity. Therefore, $iT_{SCM}$ cells may be used as an innovative strategy in the adoptive immunotherapy of cancer and infectious diseases.

## Results

**Induction of naive-like T cells from activated $CD4^+$ T cells.** The GSK-3β inhibitor TWS119 has been used for $T_{SCM}$ cell generation from naive T cells by arresting effector T cell differentiation[13,14]. This method successfully worked for naive $CD8^+$ T cells, but it was not effective for naive and differentiated $CD4^+$ T cells (Supplementary Fig. 1a–c). Therefore, we explored the signalling molecules that induce $CD4^+$ $T_{SCM}$ cells, and we subsequently discovered that Notch signalling converted activated $CD4^+$ T cells into $T_{SCM}$-like cells, which expressed naive markers, $CD44^{lo}CD62L^{hi}$. Notch signalling is important not only for T cell differentiation and function[15–17], but also for the maintenance of memory T cells *in vivo*[18]. We found that both $CD4^+$ and $CD8^+$ T cells came to express mainly Notch1 and Notch2, when they were activated under various culture conditions (Supplementary Fig. 2a,b). The method we have established is described as follows. First, to obtain antigen-specific, activated $CD4^+$ T cells, OVA-specific $CD44^{lo}CD62L^{hi}$ naive $CD4^+$ T cells were isolated from OT-II mice, and then cocultured with OVA peptide-pulsed dendritic cells (OVA-DCs). Four days later, $CD44^{hi}$ activated T cells, which showed memory precursor cell phenotypes ($CD127^+KLRG1^-$) (Supplementary Fig. 2c), were isolated by FACS and then cocultured with Notch ligand Delta-like 1-expressing murine bone marrow stromal cells (OP9-DL1 cells) or OP9 cells in the presence of IL-7 and anti-IFN-γ neutralizing antibody (Ab) for up to 12 days (Fig. 1a). When we used anti-CD3/CD28 antibodies to activate wild-type $CD4^+$ T cells *in vitro*, $CD44^{lo}CD62L^{hi}$ fraction appeared 4 days after coculture with OP9-DL1 feeder cells.

Coculturing with OP9-DL1 cells significantly induced the expression of multiple Notch target genes (*Myc, Deltex1, Hes1*) in the $CD4^+$ T cells (Supplementary Fig. 2d). Consequently, coculturing with OP9-DL1 cells in the presence of both IL-7 and anti-IFN-γ Ab let $CD4^+$ T cells survive and expand, and 20–30% of the $CD4^+$ T cells cocultured with OP9-DL1 cells downregulated CD44 expression, resulting in phenotypically naive T cells, which we called induced-$T_{SCM}$ ($iT_{SCM}$) cells and 40–60% of the $CD4^+$ T cells were $CD44^{hi}CD62L^{hi}$ T cells, which we called induced-$T_{CM}$ ($iT_{CM}$) cells (Fig. 1b). $CD44^{hi}CD62L^{lo}$ naive-like $iT_{SCM}$ cells were not induced by the coculture with OP9 cells not expressing DL1 (Fig. 1b). Both IL-7 and anti-IFN-γ Ab were essential to obtain fine proliferation of T cells on the OP9-DL1 feeder cells (Supplementary Fig. 2e). We obtained $4–5 × 10^6$ of $iT_{SCM}$ and $8–9 × 10^6$ of the $iT_{CM}$ cells from $1 × 10^6$ of the naive T cells, indicating that a portion of the activated $CD4^+$ T cells proliferated and converted into $CD44^{lo}CD62L^{hi}$ $iT_{SCM}$ cells (Fig. 1c). Activated $CD4^+$ T cells did not proliferate well during the coculture with OP9 cells, thus the number of $CD62L^+$ cells did not increase during coculture (Fig.1c). Thus, Notch signalling is necessary not only for induction of $iT_{SCM}$ cells but also for proliferation and/or survival of T cells.

To show an essential role of Notch signalling in $iT_{SCM}$ induction, the Notch inhibitor, DAPT was included in the coculture. Treatment with DAPT significantly suppressed the induction of $CD44^{lo}CD62L^{hi}$ $iT_{SCM}$ cells (Fig. 1d). Conversely, a Notch ligand-Fc protein (mDLL1-Fc) and overexpression of the Notch intracellular domain strongly induced $CD44^{lo}CD62L^{hi}$ $iT_{SCM}$ cells (Fig. 1e,f). To observe the induction of $iT_{SCM}$ cells from activated $CD4^+$ T cells at a single cell level, we first sorted IFN-γ-highly producing activated $CD4^+$ T cells (Supplementary

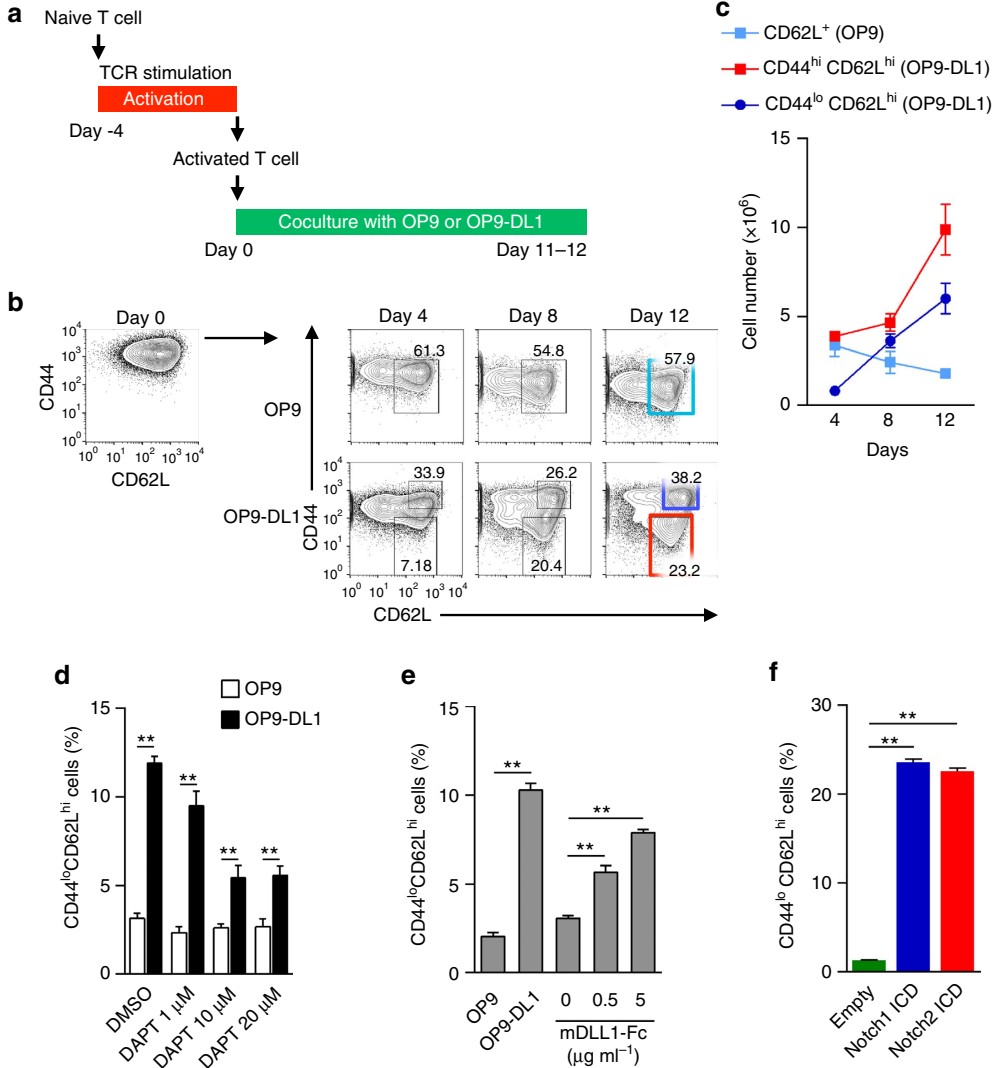

**Figure 1 | Notch signalling generates CD4$^+$ iT$_{SCM}$ cells.** (**a**) Scheme of the experimental design for the OP9 or OP9-DL1 cell coculture system. (**b**) CD44 and CD62L expression on OT-II CD4$^+$ T cells at each time point during the coculture with OP9 (upper) or OP9-DL1 (lower) cells. CD62L$^+$ cells induced by coculturing with OP9 cells are indicated in a sky-blue square (upper). Naive T cells ($1 \times 10^6$) were activated with OVA-DCs, then FACS-sorted CD44$^{hi}$ activated T cells ($1–1.5 \times 10^6$) were cocultured with OP9 or OP9-DL1 cells. CD44$^{hi}$CD62L$^{hi}$ iT$_{CM}$ and CD44$^{lo}$CD62L$^{hi}$ naive-like iT$_{SCM}$ cells induced by OP9-DL1 cells are indicated in the blue and red squares, respectively (lower). Representative data of 4 independent experiments are shown. (**c**) The cell number of gated cell population at each time point in **b** ($n = 3$). (**d**) Effects of a γ-secretase inhibitor DAPT on the induction of CD4$^+$ iT$_{SCM}$ cells. Wild-type activated CD4$^+$ T cells were cocultured with OP9 or OP9-DL1 cells with or without the presence of DAPT for four days ($n = 3$ per group). (**e**) Effects of a Notch ligand Fc protein (mDLL1-Fc) on the induction of CD4$^+$ iT$_{SCM}$ cells. Wild-type activated CD4$^+$ T cells were cultured with plate-bound mDLL1-Fc for four days ($n = 3$ per group). (**f**) Effects of overexpression of Notch ICDs on the induction of CD4$^+$ iT$_{SCM}$ cells ($n = 3$ per group). Notch1 or Notch2 ICDs were retrovirally transduced into CD4$^+$ T cells upon TCR stimulation. Notch1 or Notch2 ICD-transduced T cells were sorted as CD4$^+$CD44$^{hi}$Thy1.1$^+$ cells. Sorted cells were consequently cocultured with OP9 cells for four days. **$P < 0.01$ (Student's $t$-test). Data are representative of at least two independent experiments. Error bars show s.e.m.

Fig. 3), and then cocultured the sorted single CD4$^+$ T cells with OP9-DL1 or OP9 cells. As a result, whereas all the clones cocultured with OP9-DL1 cells produced CD44$^{lo}$CD62L$^{hi}$ iT$_{SCM}$ cells, none of the clones with OP9 cells did (Supplementary Fig. 3). These results indicated that Notch signalling induced phenotypically naive-like T cells from activated CD4$^+$ T cells.

**Characterization of CD4$^+$ iT$_{SCM}$ cells.** The cell size of iT$_{SCM}$ cells were smaller than that of activated CD4$^+$ T cells and iT$_{CM}$ cells and close to that of naive T cells (Fig. 2a). Stem-like cells have been shown to express drug transporter including

ATP-binding cassette sub-family G member 2 (ABCG2), which rapidly efflux lipophilic fluorescent dyes[19], thus exhibit so called 'side population (SP)' fraction in flow cytometry. CD4$^+$ iT$_{SCM}$ cells showed more SP cells than CD4$^+$ iT$_{CM}$ T cells did (Fig. 2b), and this fraction was decreased by the ABCG2 inhibitor Fumitremorgin C (FTC). These data suggest that CD4$^+$ iT$_{SCM}$ cells may have a characteristic feature related to stem cells.

FACS analysis revealed that the coculturing with OP9-DL1 cells suppressed the expressions of exhausted and/or anergic T cell markers, PD-1 and CTLA-4, but strongly induced Bcl-2 expression (Fig. 2c,d). Notably, the expression level of CCR7 on iT$_{SCM}$ cells was comparable to that on the original naive

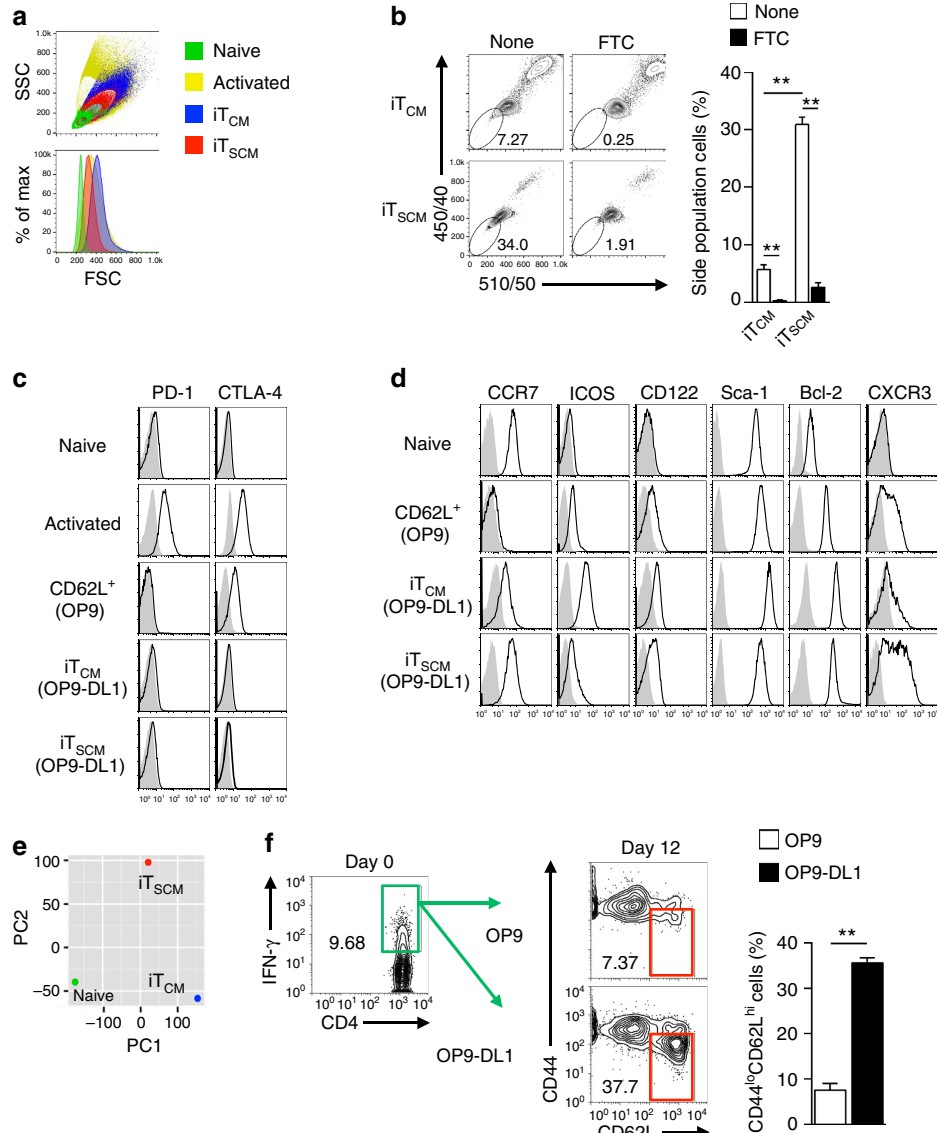

**Figure 2 | Characterization of CD4$^+$ iT$_{SCM}$ cells.** (**a**) Flow cytometry analysis for cell sizes. Cell sizes are evaluated as relative values of forward scatter (FSC). Representative data of 3 independent experiments are shown. (**b**) SP assays were performed as described in the Methods section. Representative dot plots (left). The bar graphs show the percentages of SP cells ($n=3$) (right). (**c,d**) Flow cytometry analysis for the expression of surface markers and intracellular Bcl-2 in naive CD4$^+$ T, CD62L$^+$ CD4$^+$ T, CD4$^+$ iT$_{CM}$ and CD4$^+$ iT$_{SCM}$ cells. (**e**) Principal component analysis of naive CD4$^+$ T, CD4$^+$ iT$_{CM}$, and CD4$^+$ naive-like T cells (iT$_{SCM}$). (**f**) Induction of CD4$^+$ iT$_{SCM}$ cells from *in vivo*-primed Th1 cells by coculturing with OP9 or OP9-DL1 cells. CD4$^+$ T cells isolated from immunized CD45.2$^+$ OT-II *Ifng*$^{Venus}$ mice were transferred into CD45.1$^+$ congenic mice, followed by immunization with OVA/CFA. Six days later, CD45.2$^+$ IFN-γ$^+$CD44$^{hi}$CD4$^+$ T cells (green square) were purified, and then cocultured with OP9-DL1 cells for 12 days. The iT$_{SCM}$-like gate is indicated by the red square ($n=3$ per each groups). **P<0.01 (Student's *t*-test). Data are representative of two independent experiments (**a–d,f**) and microarray data are acquired by single experiment (**e**). Error bars show s.e.m.

CD4$^+$ T cells (Fig. 2d). Compared with the CD44$^{hi}$CD62L$^{hi}$ iT$_{CM}$ cells, the CD44$^{lo}$CD62L$^{hi}$ iT$_{SCM}$ cells expressed lower levels of ICOS and a higher level of CXCR3, which are characteristic markers of T$_{SCM}$ and T$_{MNP}$ cells[10,11], and these cells also showed CD122$^{hi}$Sca-1$^{hi}$Bcl-2$^{hi}$ T$_{SCM}$-like phenotypes[12] (Fig. 2d). On the other hand, the CD62L-positive T cells (designated as CD62L$^+$ T cells) induced by coculture with OP9 control feeder cells expressed higher levels of CTLA-4 and lower levels of CCR7 and Bcl-2 compared with iT$_{CM}$ and iT$_{SCM}$ cells (Fig. 2c,d). Furthermore, surface phenotypes of CD44$^{lo}$ cells in CD62L$^+$ T cells were identical to those of CD44$^{hi}$CD62L$^+$ cells (Supplementary Fig. 4). Thus, CD62L$^+$ T cells induced by coculture with OP9 cells are phenotypically different from iT$_{SCM}$ and iT$_{CM}$ cells.

Principal component analysis and clustering analysis indicated that a gene expression pattern of the CD44$^{lo}$CD62L$^{hi}$ iT$_{SCM}$ cells differed from that of both the original naive CD4$^+$ T cells and the CD44$^{hi}$CD62L$^{hi}$ iT$_{CM}$ cells (Fig. 2e and Supplementary Fig. 5a). Expression levels of various cytokines and transcription factors were very different among naive, iT$_{CM}$ and iT$_{SCM}$ cells (Supplementary Fig. 5b). Collectively, these results indicated that Notch signal-induced naive-like CD4$^+$ T cells (CD4$^+$ iT$_{SCM}$ cells) were a unique subset with stem cell memory T cell characteristics.

Coculturing with OP9-DL1 cells induced CD4$^+$ iT$_{SCM}$ cells not only from Th1 cells but also from Th2 and Th17 cells (Supplementary Fig. 6a). CD4$^+$ iT$_{SCM}$ cells derived from Th1

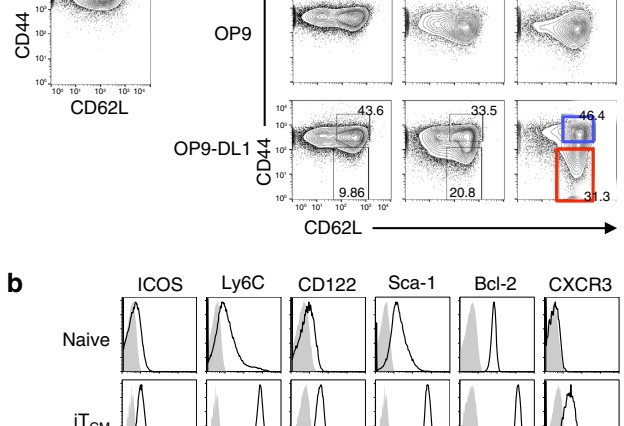

**Figure 3 | Induction of CD8$^+$ iT$_{SCM}$ cells via Notch signalling. (a)** CD44 and CD62L expression on OT-I CD8$^+$ T cells at each time point during the coculture with OP9 (upper) or OP9-DL1 (lower) cells. CD8$^+$ iT$_{CM}$ and iT$_{SCM}$ cells are indicated by blue and red squares, respectively. (**b**) Flow cytometry analysis for the expression of surface markers and intracellular Bcl-2 in naive CD8$^+$ T, CD8$^+$ iT$_{CM}$ and CD8$^+$ iT$_{SCM}$ cells. Representative data of three independent experiments are shown in (**a**) and (**b**).

cells produced a large number of IFN-$\gamma^+$ cells, when restimulated with OVA-DCs under any Th-polarizing conditions in vitro (Supplementary Fig. 6b). Similarly, only IFN-$\gamma^+$ effector cells were detected in vivo when Th1 cell-derived iT$_{SCM}$ or iT$_{CM}$ cells were administered into OT-II mice then immunized with OVA/incomplete Freund's adjuvant (IFA)(Supplementary Fig. 6c), suggesting that the cytokine-producing profiles are imprinted in CD4$^+$ iT$_{SCM}$ cells from the original Th subsets. We also found that the OP9-DL1 cell coculture system generated CD4$^+$ iT$_{SCM}$ cells from in vivo-induced Th1 cells that were derived from IFN-$\gamma$ reporter OT-II mice (Ifng$^{Venus}$) immunized with OVA/complete Freund's adjuvant (CFA) (Fig. 2f).

**Induction of CD8$^+$ iT$_{SCM}$ cells by Notch signalling.** Next, we attempted to induce CD8$^+$ iT$_{SCM}$ cells using the OP9-DL1 cell coculture system. First, we activated OVA-specific CD44$^{lo}$CD62L$^{hi}$ naive CD8$^+$ T cells from OT-I mice by coculturing with OVA-DCs. Four days later, CD44$^{hi}$ activated T cells, which expressed Notch1 and Notch2, but not Notch3, and showed the memory precursor cell phenotypes (Supplementary Fig. 2a,c), were cocultured with OP9-DL1 or OP9 cells in the presence of IL-7 and anti-IFN-$\gamma$ neutralizing Ab for 11 days. As a result, CD44$^{lo}$CD62L$^{hi}$ naive-like T cells were induced by coculturing with OP9-DL1, but not OP9, cells (Fig. 3a). Coculturing with OP9-DL1 cells upregulated CD122, Sca-1 and Bcl-2 expression on both CD44$^{lo}$CD62L$^{hi}$ naive-like iT$_{SCM}$ cells and CD44$^{hi}$CD62L$^{hi}$ iT$_{CM}$ cells (Fig. 3b). However, CD8$^+$ iT$_{SCM}$ cells expressed lower levels of ICOS and Ly6C, and a higher level of CXCR3, compared with the iT$_{CM}$ cells (Fig. 3b). These results indicated that coculture with OP9-DL1 cells induces iT$_{SCM}$ cells from activated CD8$^+$ T cells as well.

**Stem cell memory properties of antigen-specific iT$_{SCM}$ cells.** Then, we examined whether iT$_{SCM}$ cells actually possess functional stem cell memory-like phenotypes. First, cell division in response to TCR stimulation was compared among naive CD4$^+$ T cells, in vivo CD4$^+$ T$_{EM}$ and T$_{CM}$ cells, in vitro-generated Th1 cells, and Th1-derived iT$_{CM}$ and iT$_{SCM}$ cells. All these T cells were prepared from Rag2$^{-/-}$ OT-II mice, therefore, uniformly carried OVA-specific TCR. We found that CD4$^+$ iT$_{SCM}$ cells proliferated more rapidly than any of the other subsets after stimulation with OVA-DCs (Fig. 4a). OVA-specific CD8$^+$ iT$_{SCM}$ cells also highly proliferated in vitro in response to OVA restimulation compared with naive CD8$^+$ T and iT$_{CM}$ cells (Supplementary Fig. 7a). The in vitro proliferation activity of CD62L$^+$CD44$^{hi}$ and CD62L$^+$CD44$^{lo}$ T cells induced by OP9 cell coculture was inferior to that of iT$_{SCM}$ cells (Supplementary Fig. 7b). We also found that antigen-activated iT$_{SCM}$ cells retained rapid proliferation potential in the secondary coculture with OP9-DL1 cells (Supplementary Fig. 7c).

To investigate their short term proliferation capacity in vivo, we transferred each of the CFSE-labelled congenically marked CD45.1$^+$CD4$^+$ OT-II T cell subsets into CD45.2$^+$ wild-type mice, then immunized the mice with OVA. Three and six days later, the numbers of proliferated T cells derived from CD4$^+$ iT$_{SCM}$ cells in the secondary lymphoid organs were significantly greater than those from any of the other subsets (Fig. 4b for three days and Supplementary Fig. 7d,e for six days).

To assess the proliferation capacity under homoeostatic conditions in vivo, CFSE-labelled naive T cells, T$_{CM}$ cells from immunized mice, iT$_{CM}$ or iT$_{SCM}$ cells from Rag2$^{-/-}$ OT-II mice were transferred into sublethally irradiated congenic mice. After 20 days, a larger number of iT$_{SCM}$ cells were recovered, and about 70% of the T cells derived from the CD4$^+$ iT$_{SCM}$ cells retained CD44$^{lo}$CD62L$^{hi}$ T$_{SCM}$ phenotypes and divided significantly better than naive T cells, T$_{CM}$ and iT$_{CM}$ cells did (Fig. 4c). Even 150 days later, significantly greater numbers of T cells derived from the CD4$^+$ iT$_{SCM}$ cells were observed in the secondary lymphoid organs compared with those from naive CD4$^+$ T and the iT$_{CM}$ cells, and about 70% of the T cells from the CD4$^+$ iT$_{SCM}$ cells retained CD44$^{lo}$CD62L$^{hi}$ naive phenotypes (Fig. 5a for cell numbers 150 days after transplantation and Supplementary Fig. 8 for time course experiments). These results indicated that antigen-specific iT$_{SCM}$ cells possessed the great capacity not only to proliferate in response to antigen restimulation but also to survive for an extended time under homoeostatic conditions.

Next, to investigate whether CD4$^+$ iT$_{SCM}$ cells possess a self-renewal capacity in vivo, we performed serial transfer experiments; we purified CFSE$^{lo}$CD44$^{lo}$CD62L$^{hi}$ CD4$^+$ T cells derived from naive CD4$^+$ T or CD4$^+$ iT$_{SCM}$ cells from the first transferred mice, transferred them into the second mice and performed the same procedures for the third mice. In the case of CD4$^+$ iT$_{CM}$ cells, we purified and transferred CFSE$^{lo}$CD44$^{hi}$CD62L$^{hi}$ CD4$^+$ T cells. CD4$^+$ T cells derived from the CD4$^+$ iT$_{SCM}$ cells were successfully detected in the second lymphoid organs of the third transferred mice, while naive CD4$^+$ T and CD4$^+$ iT$_{CM}$ cells failed to be detected in the second and third transferred mice, respectively (Fig. 5b), indicating that CD4$^+$ iT$_{SCM}$ cells possess high self-renewal capacity in vivo. Collectively, these results suggested that CD4$^+$ iT$_{SCM}$ cells had stem cell-like potential.

**Reduced expression of p53 in iT$_{SCM}$ cells.** Then, we investigated the mechanism of a high proliferation potential of iT$_{SCM}$ cells. Rapid proliferation and the extended survival of CD4$^+$ iT$_{SCM}$ cells was not due to high expression of IL-2 or higher sensitivity

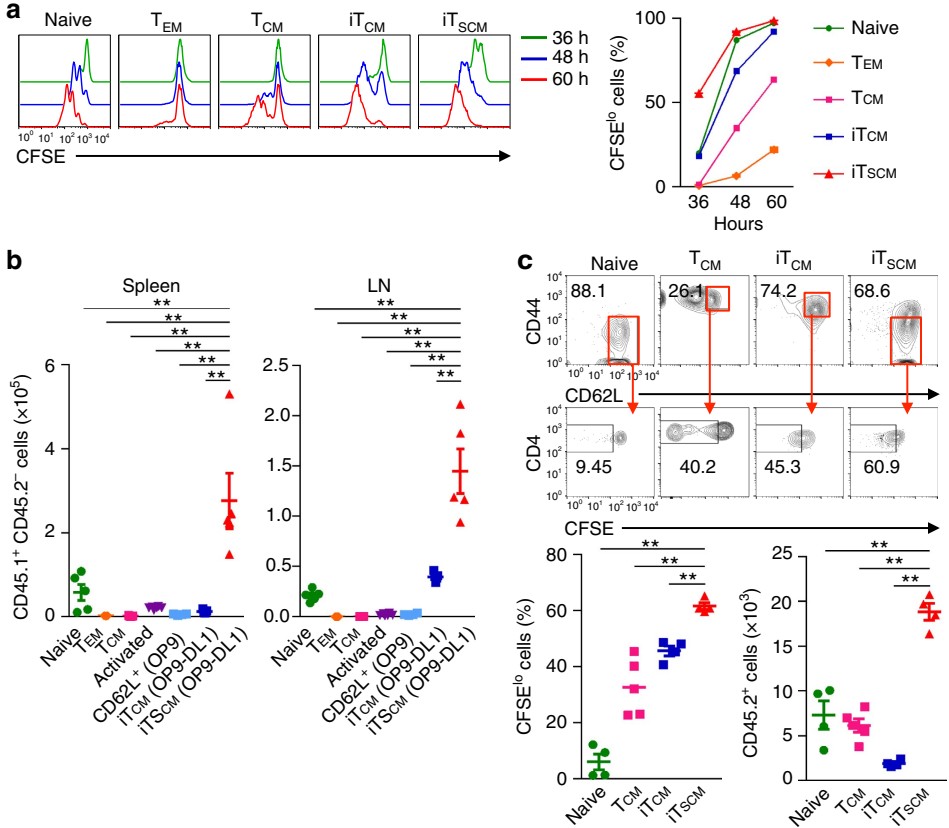

**Figure 4 | Murine CD4$^+$ iT$_{SCM}$ cells possess high proliferation ability.** (**a**) Cell division of CD4$^+$ T cells after stimulation with OVA-DCs *in vitro* assessed by CFSE-dilution assay. Naive CD4$^+$ T cells, T$_{EM}$, and T$_{CM}$ cells were isolated from immunized *Rag2$^{-/-}$* OT-II mice, and iT$_{CM}$, and iT$_{SCM}$ cells were generated from *in vitro* activated *Rag2$^{-/-}$* OT-II CD4$^+$ T cells. These cells were CFSE labelled, then stimulated with OVA-DCs for indicated periods. CFSE profiles and the fraction of CFSE$^{lo}$ proliferated cells are shown (*n* = 3). (**b**) *In vivo* expansion of iT$_{SCM}$ and other type of CD4$^+$ T cells after antigen restimulation. CD45.1$^+$CD4$^+$OT-II naive T cells, T$_{EM}$, T$_{CM}$ cells, CD62L$^+$ cells, iT$_{CM}$ and iT$_{SCM}$ (2 × 10$^5$) were transferred into CD45.2$^+$ mice, followed by OVA/IFA immunization. Three days later, cell numbers in the spleen and LNs were counted (*n* = 5 per group). (**c**) *In vivo* expansion of CD4$^+$ iT$_{SCM}$ cells under homoeostatic conditions. CFSE-labelled 5 × 10$^5$ cells were injected into the sublethally irradiated recipient mice, then 20 days later, T cells were isolated and analysed CD44/CD62L expression and CFSE profiles. The percentage of CFSE$^{lo}$CD44$^{lo}$CD62L$^{hi}$ fraction for iT$_{SCM}$ cells or naive T cells, and that of CFSE$^{lo}$CD44$^{hi}$CD62L$^{hi}$ fraction for T$_{CM}$ and iT$_{CM}$ cells as well as the total cell numbers recovered from the spleen and LNs are shown. **$P$ < 0.01 (one-way ANOVA (**a–c**)). Data are representative of at least two independent experiments. Error bars show s.e.m.

to IL-2: OVA-specific CD4$^+$ iT$_{SCM}$ cells produced less but sufficient amounts of IL-2 after stimulation with OVA-DCs (Supplementary Fig. 9a). In addition, there were no differences in the phosphorylation levels of Stat5 and Akt, which are downstream of the IL-2 receptor signalling pathway, between CD4$^+$ iT$_{SCM}$ and iT$_{CM}$ cells (Supplementary Fig. 9b). We noticed that CD4$^+$ iT$_{SCM}$ cells composed a higher proportion of G2/M and S phases even on Day 1 after stimulation compared with CD4$^+$ iT$_{CM}$ cells and naive CD4$^+$ T cells, which required 2-3 days to enter G2/M and S phases (Supplementary Fig. 10a). We also found that apoptosis of CD4$^+$ iT$_{SCM}$ cells after TCR restimulation was significantly lower than CD4$^+$ iT$_{CM}$ cells did (Supplementary Fig. 10b).

The tumour suppressor gene p53 is known to be a regulatory factor in cell cycle arrest and apoptosis[20]. It has been proposed the multilevel crosstalk between the Notch pathway and the p53 pathways[21]. The Notch signal has been shown to suppress p53 expression in T cell lymphoma cells[22]. p53 expression levels in CD4$^+$ iT$_{SCM}$ cells were significantly lower than that of CD4$^+$ naive and iT$_{CM}$ cells *in vitro* and *in vivo* (Supplementary Fig. 10c,d). Consistently, CD4$^+$ iT$_{SCM}$ cells expressed significantly lower levels of p53 target genes: a p53-specific E3 ubiquitin ligase *Mdm2*, cell growth arrest genes *Gadd45a* and

*Ptprv*, and apoptosis regulation genes *Bax*, *Bbc3* and *Fas* (Supplementary Fig. 10e). Degradation of p53 by Mdm2 has been reported to be critical for CD4$^+$ T cell proliferation on TCR stimulation[23]. Mdm2 inhibitor Nutlin3a decreased the G2/M phase proportion and increased AnnexinV$^+$PI$^+$ apoptotic cells (Supplementary Fig. 10f), suggesting that p53 reduction is involved in higher proliferation and lower apoptotic potentials of iT$_{SCM}$ cells. CD8$^+$ iT$_{SCM}$ cells also composed a higher proportion of G2/M and S phases and fewer AnnexinV$^+$PI$^+$ apoptotic cells than CD8$^+$ iT$_{CM}$ cells did (Supplementary Fig. 11a,b), which is related to a lower expression of p53 and its target genes than in CD8$^+$ iT$_{CM}$ cells (Supplementary Fig. 11c,d). These results suggested that lower expression levels of p53 and its targets molecules in iT$_{SCM}$ cells could be a mechanism for iT$_{SCM}$ cells to be resistant to cell cycle arrest and apoptosis after TCR stimulation.

**Potent antitumour activity of iT$_{SCM}$ cells.** iT$_{SCM}$ cells showed strong proliferation upon TCR restimulation with resistance to cell cycle arrest and apoptosis, which suggested that iT$_{SCM}$ cells were more likely to be applicable to cancer immunotherapy[8,24]. Thus, we investigated the antitumour effects of iT$_{SCM}$ cells using

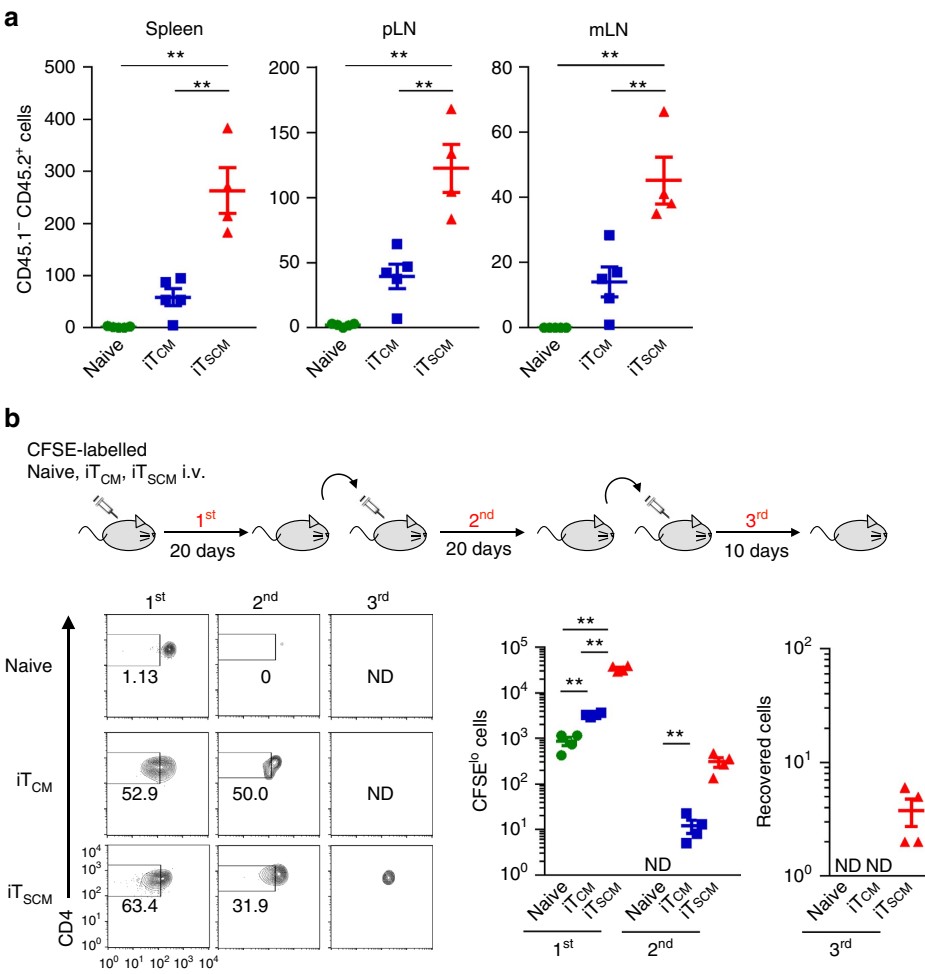

**Figure 5 | Murine CD4$^+$ iT$_{SCM}$ cells possess long-term survival ability.** (**a**) Long-term survival assay *in vivo*. *Rag2$^{-/-}$* OT-II naive T, iT$_{CM}$ and iT$_{SCM}$ cells (5 × 10$^4$) were adoptively transferred into sublethally irradiated CD45.1$^+$ congenic mice. After 150 days, numbers of the CD45.2$^+$ T cells were determined (*n* = 5 for naive T and iT$_{CM}$; *n* = 4 for iT$_{SCM}$). (**b**) Serial transfer of *Rag2$^{-/-}$* OT-II CD45.2$^+$CD4$^+$ naive T, iT$_{CM}$ and iT$_{SCM}$ cells. CFSE-labelled 1 × 10$^6$ T cells were injected into the CD45.1$^+$ recipient mice (first transfer). On day 20, CFSE$^{lo}$CD44$^{lo}$CD62L$^{hi}$ fraction for iT$_{SCM}$ cells or naive T cells, or CFSE$^{lo}$CD44$^{hi}$CD62L$^{hi}$ fraction for iT$_{CM}$ cells were sorted and re-labelled with CFSE, and then transferred into the irradiated CD45.1$^+$ recipient mice (second transfer). Twenty days later, the same procedure was performed as third transfer, and mice were analysed on day 10 after third transfer. CFSE profiles in CD44$^{lo}$CD62L$^{hi}$ fraction for iT$_{SCM}$ cells or naive T cells, or CD44$^{hi}$CD62L$^{hi}$ fraction for iT$_{CM}$ cells are shown in left panels. The actual numbers of the recovered CFSE$^{lo}$ cells after first and second transfer and the number of whole cells recovered after third transfer are shown in right graphs (*n* = 5 per group). ND, not detectable. **P < 0.01 (one-way ANOVA (**a**,**b**)). Data are representative of at least two independent experiments. Error bars show s.e.m.

T cell lymphoma cell line E.G7-OVA-bearing mice. First we transferred CD8$^+$ iT$_{CM}$ and iT$_{SCM}$ cells derived from OT-I mice into E.G7-OVA-bearing mice. After 36 hours, CD8$^+$ iT$_{SCM}$ cells were detected in the sentinel lymph nodes, whereas much fewer CD8$^+$ iT$_{CM}$ cells were present (Fig. 6a). OVA-specific CD8$^+$ iT$_{SCM}$ cells also proliferated faster in E.G7-OVA-bearing mice in response to OVA restimulation than OVA-specific iT$_{SCM}$ cells did (Supplementary Fig. 12a). It has been reported that naive T cells are appropriate for cancer immunotherapy because of their stronger replicative potential compared to that of any other memory subset[8]. On the other hand, activated T cells that acquire effector functions *in vitro* presumably lose the antitumour efficacy *in vivo*[4]. Thus, we compared the antitumour effects against E.G7-OVA among naive CD8$^+$ T, OVA-activated CD8$^+$ T and CD8$^+$ iT$_{SCM}$ cells that were all derived from OT-I mice. As shown in Fig. 6b, OVA-specific CD8$^+$ iT$_{SCM}$ cells showed significantly stronger suppressive effects on E.G7-OVA cell

growth than OVA-specific naive CD8$^+$ T and activated CD8$^+$ T cells did. Consequently, OVA-specific CD8$^+$ iT$_{SCM}$ cells improved the survival rates of the mice (Fig. 6b). Similarly, OVA-specific CD4$^+$ iT$_{SCM}$ cells also survived in the sentinel lymph nodes and suppressed E.G7-OVA cell growth more efficiently than CD4$^+$ T$_{EM}$ and T$_{CM}$ cells, iT$_{CM}$ cells, and CD62L$^+$ cells, although CD4$^+$ iT$_{SCM}$ cells showed weaker tumour suppressing effects than CD8$^+$ iT$_{SCM}$ cells did (Supplementary Fig. 12b,c).

To investigate whether antigen-specific CD4$^+$ iT$_{CM}$ or iT$_{SCM}$ cells enhance the antitumour effects of the CD8$^+$ iT$_{SCM}$ cells, we transferred various combinations of OVA-specific CD4$^+$ and CD8$^+$ iT$_{CM}$ or iT$_{SCM}$ cells into E.G7-OVA-bearing mice. The CD8$^+$ iT$_{SCM}$ cells exerted not only stronger suppressive effects against the E.G7-OVA cell growth but also better survival rates of the mice in combination with the CD4$^+$ iT$_{SCM}$ cells than alone or in combination with the CD4$^+$ iT$_{CM}$ cells (Fig. 6c,d).

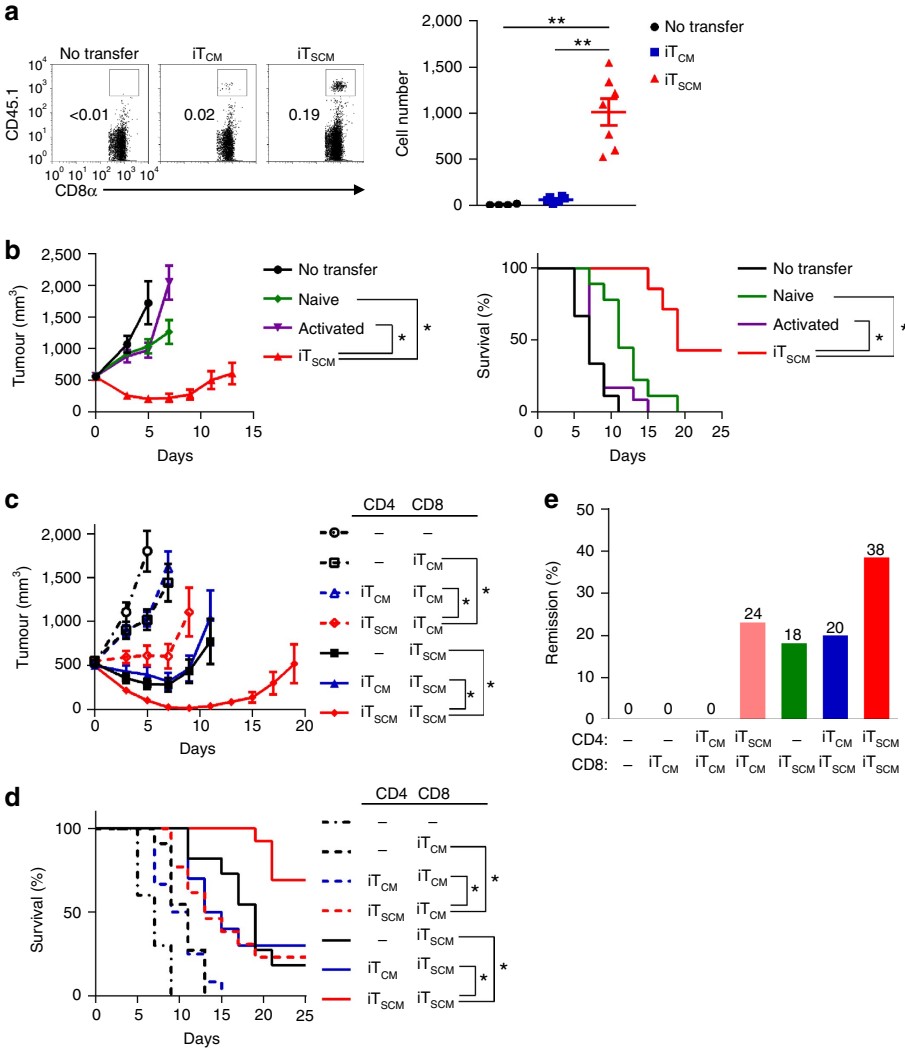

**Figure 6 | Antitumour potential of murine iT$_{SCM}$ cells. (a)** Flow cytometry analysis of OVA-specific CD8$^+$ T cells infiltrating into sentinel LNs. CD45.2$^+$ wild-type mice were inoculated with E.G7-OVA eight days before the transfer of $1.5 \times 10^5$ CD45.1$^+$ OT-I CD8$^+$ iT$_{CM}$ or iT$_{SCM}$ cells. T cells in the sentinel LNs were analysed by flow cytometry 36 h after the T cell transfer. Squares in the dot plots indicate CD45.1$^+$ T cells (left). ($n = 4$ for no transfer; $n = 6$ for iT$_{CM}$ cells; $n = 7$ for iT$_{SCM}$ cells). **(b)** Tumour volumes (left) and survival rates (right) of E.G7-OVA-bearing mice. $3 \times 10^5$ naive CD8$^+$ T, CD44$^{hi}$ activated CD8$^+$ T, or CD8$^+$ iT$_{SCM}$ cells derived from OT-I mice were adoptively transferred into the E.G7-OVA-bearing mice ($n = 9$ for no transfer and naive; $n = 12$ for activated; $n = 7$ for iT$_{SCM}$). **(c–e)** The effect of combination of OT-I CD8$^+$ iT$_{CM}$ or iT$_{SCM}$, and OT-II CD4$^+$ iT$_{CM}$ or iT$_{SCM}$ cells on antitumour immunity. T cells ($3 \times 10^5$ cells for each group) were adoptively transferred into E.G7-OVA-bearing mice. **(c)** Tumour volumes of E.G7-OVA-bearing mice. **(d)** Survival rates of E.G7-OVA-bearing mice. **(e)** Remission rates at 90 days after the adoptive T cell transfer. Data were collected from three independent experiments ($n = 10$ for no transfer; $n = 11$ for CD8$^+$ iT$_{CM}$; $n = 12$ for CD4$^+$ iT$_{CM}$ and CD8$^+$ iT$_{CM}$; $n = 13$ for CD4$^+$ iT$_{SCM}$ and CD8$^+$ iT$_{CM}$; $n = 11$ for CD8$^+$ iT$_{SCM}$; $n = 10$ for CD4$^+$ iT$_{CM}$ and CD8$^+$ iT$_{SCM}$; $n = 13$ for CD4$^+$ iT$_{SCM}$ and CD8$^+$ iT$_{SCM}$) **(c,d,e)**. *$P < 0.05$, **$P < 0.01$ (one-way ANOVA (**a**), Student's $t$-test (**b** left, **c** left), and the Kaplan–Meier method (**b** right, **d**)). Data are representative of at least two independent experiments. Error bars show s.e.m.

Consistent with the results of Fig. 6a, OVA-specific CD8$^+$ iT$_{CM}$ cells alone had some tumour-suppressive effects, however, combination with OVA-specific CD4$^+$ iT$_{SCM}$ significantly improved the antitumour effects (Fig. 6c,d), and their combination induced remission in 24% of the mice (Fig. 6e). More importantly, the remission rate by OVA-specific CD8$^+$ iT$_{SCM}$ cells alone (18%) was substantially improved to 38% by the combination with the CD4$^+$ iT$_{SCM}$ cells (Fig. 6e). These results suggested that antigen-specific CD4$^+$ iT$_{SCM}$ cells enhanced antitumour immune responses provided by antigen-specific CD8$^+$ iT$_{SCM}$ cells. Taken together, tumour antigen-specific CD8$^+$ iT$_{SCM}$ cells are more likely to have potent antitumour effects, and the combination with the CD4$^+$ iT$_{SCM}$ cells and the CD8$^+$ iT$_{SCM}$ cells are thought to provide the strongest antitumour effects.

**Comparison between Wnt and Notch-induced T$_{SCM}$-like cells.** We next compared the antitumour effects of Wnt signalling-induced T$_{SCM}$ cells and Notch signalling-induced iT$_{SCM}$ cells in mouse models. As shown previously[12], TCR stimulation in the presence of TWS119, a Wnt activator, generated OVA-specific CD44$^{lo}$CD62L$^{hi}$ CD8$^+$ T$_{SCM}$-like cells (Supplementary Fig. 13a). The number of iT$_{SCM}$ cells generated by OP9-DL1 coculture was much higher than that of TWS119-induced T$_{SCM}$-like cells from the same number of naive CD8$^+$ T cells (Supplementary Fig. 13b). This is probably due to a strong anti-proliferative effect of TWS119. When we injected the same number of OVA-specific iT$_{SCM}$ cells or TWS119-induced T$_{SCM}$-like cells into E.G7-OVA-bearing mice, both types of T cells showed similar antitumour effects. These results suggest that the antitumour efficacy of iT$_{SCM}$ cells is

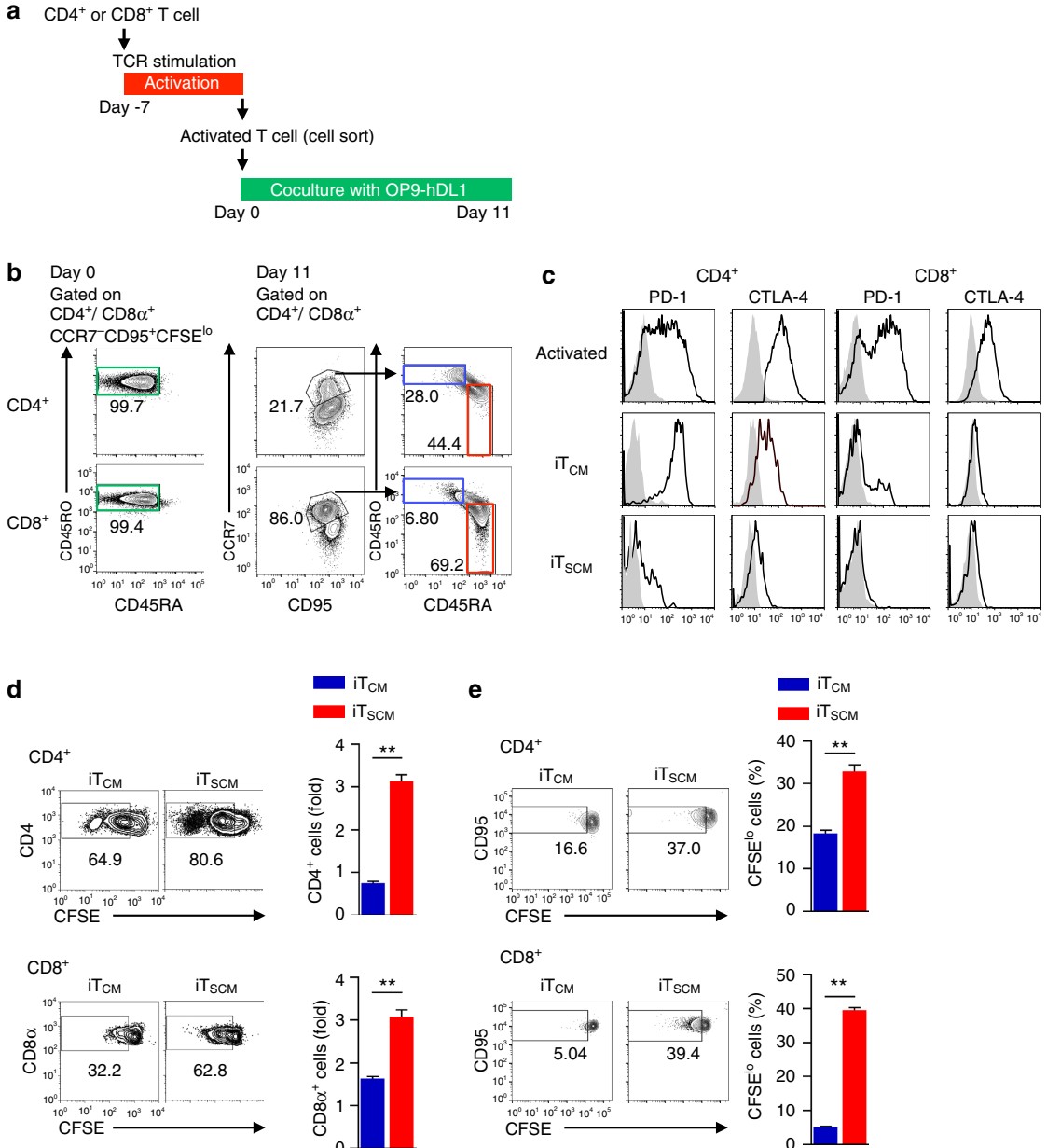

**Figure 7 | Generation of human CD4$^+$ and CD8$^+$ iT$_{SCM}$ cells.** (**a**) Scheme for inducing antigen-specific human CD4$^+$ and CD8$^+$ iT$_{SCM}$ cells. CD4$^+$ or CD8$^+$ T cells from PBMC were activated with EBV-transformed autologous LCL then cocultured with OP9-hDL1 cells. (**b**) Generating EBV-specific CD4$^+$ and CD8$^+$ iT$_{SCM}$ cells. CFSE-labelled peripheral CD4$^+$ or CD8$^+$ memory (CD45RA$^-$CD45RO$^+$CCR7$\pm$CD95$^+$) T cells were cocultured with 40 Gy irradiated EBV-transformed autologous LCL for seven days. EBV-specific activated T cells (CD45RA$^-$CD45RO$^+$CCR7$^-$CD95$^+$CFSE$^{lo}$) (Day 0) were sorted, and then cocultured with OP9-hDL1 cells for 11 days. Cells on Day 0 and 11 were subjected to flow cytometry analysis. CD45RA/CD45RO profiles of activated T cells on Day 0, CCR7/CD95 profiles on Day 11 after coculture with OP9-hDL1, and CD45RA/CD45RO profiles of the CCR7$^+$CD95$^+$ fraction are shown. Representative data of three-independent experiments are shown. (**c**) Expression of PD-1 and CTLA-4 on human activated T, iT$_{CM}$ and iT$_{SCM}$ cells. Representative data of three-independent experiments are shown. (**d**) Recall responses to EBV. CFSE-labelled EBV-specific iT$_{CM}$ and iT$_{SCM}$ cells ($5 \times 10^4$) were cocultured with autologous LCL for 60 h. The percentages of CFSE$^{lo}$ cells are shown (left). The bar graphs show the fold increase of recovered T cells ($n = 3$ per group) (right). (**e**) Proliferative responses to IL-7. iT$_{CM}$ and iT$_{SCM}$ cells were cultured in the presence of IL-7 for seven days. The percentages of CFSE$^{lo}$ cells are shown (left). The bar graphs show the fraction of CFSE$^{lo}$ T cells ($n = 3$ per group) (right). **$P < 0.01$ (Student's $t$-test). Data are representative of independent experiments using human samples provided by three healthy donors. Error bars show s.e.m.

equivalent to that of TWS119-induced T$_{SCM}$-like cells; however, coculture with OP9-DL1 is advantageous in generating a larger number of antigen-specific T$_{SCM}$-like cells.

**Generation of iT$_{SCM}$ cells from human T cells.** We then attempted to generate antigen-specific iT$_{SCM}$ cells from human

peripheral blood T cells by coculturing with human DLL1-expressing OP9 (OP9-hDL1) cells. CD4$^+$ and CD8$^+$ memory T cells were cocultured with autologous Epstein Barr virus (EBV)-transformed lymphoblastoid cell lines (LCLs)[25,26]. Seven days later, EBV-specific activated T cells were sorted as CD45RA$^-$CD45RO$^+$CCR7$^-$CD95$^+$CFSE$^{lo}$ cells (Supplementary Fig. 14a) and then cocultured with OP9-hDL1 cells for

11 days (Fig. 7a). Coculturing with OP9-hDL1 cells induced $CD45RA^+CD45RO^-CCR7^+CD95^+$ $T_{SCM}$-like cells and $CD45RA^-CD45RO^+CCR7^+CD95^+$ $iT_{CM}$ cells from both EBV-specific $CD4^+$ and $CD8^+$ T cells; we defined these cells as human $iT_{SCM}$ and $iT_{CM}$ cells, respectively (Fig. 7b). To achieve substantial conversion to $T_{SCM}$ phenotypes, over 11 days coculture with OP9-hDL1 cells was necessary (Supplementary Fig. 14b). Conversion rate was varied among three independent donors (Supplementary Fig. 14b and c). Conversion efficiency from EBV-specific activated human T cells to $iT_{SCM}$ is 5–9% for $CD4^+$ T cells and 12–66% for $CD8^+$ T cells depending on the donors. IL-15 is well known as a cytokine associated with memory cell proliferation and homoeostasis. Coculturing with OP9-hDL1 cells in the presence of IL-15 strongly reduced the efficiency of $iT_{SCM}$ cell induction (Supplementary Fig. 14d). We also successfully induced melanoma antigen Mart-1-specific $CD8^+$ $iT_{SCM}$ cells from healthy donors (Supplementary Fig. 15a). Coculturing with OP9-hDL1 cells induced downregulation of PD-1 and CTLA-4 expressions on both EBV-specific $CD4^+$ and $CD8^+$ T cells. PD-1 and CTLA-4 expression levels on EBV-specific $iT_{SCM}$ cells were much lower than those on EBV-specific $iT_{CM}$ cells (Fig. 7c). To investigate the EBV-specific recall response, we cocultured EBV-specific $CD4^+$ and $CD8^+$ $iT_{CM}$ and $iT_{SCM}$ cells with autologous LCLs for 6 h. We found that EBV-specific $CD4^+$ $iT_{SCM}$ cells generated a higher percentage of $IFN-\gamma^+CD40L^+$ cells than the $CD4^+$ $iT_{CM}$ cells did. On the other hand, EBV-specific $CD8^+$ $iT_{SCM}$ cells produced less perforin- and/or granzymeB-producing cells than the $CD8^+$ $iT_{CM}$ cells did, resulting in sufficient but lower killing activity against autologous LCLs (Supplementary Fig. 15b,c). However, 60 h after coculture, EBV-specific $CD8^+$, as well as $CD4^+$ $iT_{SCM}$ cells divided better than $CD8^+$ and $CD4^+$ $iT_{CM}$ cells did. Consequently, both of the EBV-specific $iT_{SCM}$ cells recovered greater numbers of the cells than the $iT_{CM}$ cells did (Fig. 7d). In addition, $CD4^+$ and $CD8^+$ $iT_{SCM}$ cells divided more rapidly in the presence of a homoeostatic cytokine IL-7 than $CD4^+$ and $CD8^+$ $iT_{CM}$ cells did (Fig. 7e). These results indicated that the Notch signalling generated antigen-specific $iT_{SCM}$ cells from human peripheral blood $CD4^+$ and $CD8^+$ T cells.

**Comparison of human $iT_{SCM}$ cells with other memory T cells**. Last, we investigated the ontology of human $iT_{SCM}$ cells. We compared gene expression profiles of human $CD8^+$ $iT_{SCM}$ cells, $T_{SCM}$ cells reported by Gattinoni et al., and $T_{MNP}$ cells reported by Pulko et al., which were currently known as memory T cell subsets with a naive T cell phenotype[10,11,24]. The data on those memory T cells were obtained from a public database (GSE2331, GSE80306, GSE68003). For this comparison, we focused on 1771 transcripts found to be differentially expressed among naive, $T_{CM}$, $T_{EM}$, $T_{SCM}$, $T_{MNP}$ and TWS119-induced $T_{SCM}$ cells generated in vitro (called '$T_{SCM}$-enriched' in ref. 24) and our Notch-induced $iT_{SCM}$ cells. Clustering analysis revealed that $iT_{SCM}$ cells were most closely related to TWS119-induced $T_{SCM}$ cells (Supplementary Fig. 16a). Pathway analysis of 1771 selected genes also revealed that pathway profiles of $iT_{SCM}$ cell were similar to those of TWS119-induced $T_{SCM}$ cells. Importantly, the gene expression levels of pathways for oxidative phosphorylation, cell growth and interferon response were higher both in TWS119-induced $T_{SCM}$ and Notch-induced $iT_{SCM}$ cells than in other memory T cell subsets (Supplementary Fig. 16b). PC analysis also revealed that Notch-induced $iT_{SCM}$ was positioned most closely to TWS119-induced $T_{SCM}$, but was distantly related to in vivo $T_{SCM}$ and $T_{MNP}$ cells (Supplementary Fig.16c).

## Discussion

The conversion of the cellular fate of differentiated cells is a current challenge facing modern cell biology. The strategy of converting from 'exhausted' T cells into 'vigorous' T cells is important for adoptive T-cell immunotherapy because the quality of T cells decides the improvement in patient outcomes. Recent studies have challenged the rejuvenation of differentiated T cell by various approaches. Refreshment of T cells via iPS cells derived from antigen-specific T cells (T-iPS cells) is promising[5,6]. However, a significant length of time is needed to establish iPS cells, and the efficiency of generating T cells from iPS cells is still not high. Newly identified memory T cell subsets with naive markers, $T_{SCM}$ and $T_{MNP}$ cells possess stronger proliferative potential than naive and other memory cells do. The generation of $T_{SCM}$ cells is a very effective strategy for target-specific T cell therapy[10]. Transplantation of quasi naive T cells could provide the breakthrough in developing a more effective immunotherapy.

In this study, we propose a novel strategy of generating Notch signalling-induced $T_{SCM}$ cells ($iT_{SCM}$ cells) from activated T cells by coculturing with OP9 cells expressing a Notch ligand. $iT_{SCM}$ cells possessed longevity, the potential to produce a large number of effector cells and exhibited potent antitumour activities. Notably, the combinational transplantation of $CD4^+$ and $CD8^+$ $iT_{SCM}$ cells achieved approximately 40% of a complete remission state in tumour-bearing mice. In addition to the method of inducing $T_{SCM}$ cells by Wnt activators, our method can generate $T_{SCM}$-like cells from conventional memory T cells by Notch signalling. This method has an advantage compared with Wnt activators since $iT_{SCM}$ cells can be converted from antigen-specific memory T cells, thus TCR gene transduction is not necessary.

Gene expression profile analysis revealed that our Notch-induced $iT_{SCM}$ cells were most similar to TWS119-induced $T_{SCM}$ cells generated in vitro. These two in vitro-generated $T_{SCM}$-like cells were distantly related to in vivo $T_{SCM}$ and $T_{MNP}$ cells. Our data suggested that Notch-induced $iT_{SCM}$ cells were very different from other memory T cell subsets including in vivo-derived $T_{CM}$, $T_{EM}$, $T_{SCM}$ and $T_{MNP}$ cells. PC analysis suggests that TWS119-induced $T_{SCM}$ cells are also slightly distinct from in vivo memory T cells. Thus, in vitro generated $T_{SCM}$-like cells induced by Notch or Wnt signals have a specific character acquired during in vitro stimulation. Nevertheless, in vitro generated $T_{SCM}$-like cells were extremely useful for T cell based adoptive cancer immunotherapy. Although two in vitro generated $T_{SCM}$-like cells possessed a similar antitumour ability, a significantly greater number of $T_{SCM}$-like cells can be obtained by coculture with Notch-ligand expressing stroma cells compared with TWS119 treatment. Therefore, our method may have another advantage for adoptive immunotherapy in terms of the cell number.

Notch signalling has been shown to be closely linked to various immune responses, especially the formation and maintenance of memory T cells. A recent paper suggested that Notch signalling promotes the differentiation of terminal effector cells in the contracting phase; therefore, the lack of Notch1 and Notch2 expanded the mature memory T cell pool[27]. In contrast, an attribution of HIF-1α and Notch to long-term survival of effector memory Th17 cells[28], Notch signalling has been shown to be essential for the survival of conventional memory T cells by increasing glucose uptake[18]. Aberrant Notch signalling in T cell acute lymphoblastic leukaemia has been shown to induce activation of the PTEN-PI3 kinase pathway and the down-modulation of p53 (refs 21,29). These studies suggested that Notch signalling played divergent roles in the contraction and memory phase of the T cell response. Our studies indicate that Notch signalling redirects differentiation from memory precursor-like activated T cells to $iT_{SCM}$ cells. It is notable that

T cells are generated from iPS cells by the coculture with the same OP9-DL1 cells[5,6].

PD-1 and CTLA-4 have been shown to be a marker of exhausted T cells and are now important targets of antitumour therapy[30–32]. A high expression of PD-1 and CTLA-4 in activated T cells is restored to levels similar to those of naive T cells through iT$_{SCM}$ cell induction. Low levels of these inhibitory receptors may explain a rapid proliferative response to antigens and the stronger antitumour potential of iT$_{SCM}$ cells. Expression of PD-1 and CTLA-4 may not be down-regulated by Notch, since we observed similar reduction of PD-1 and CTLA-4 in T cells after coculture with OP9 not expressing Delta-like 1. The mechanism of reduction of PD-1 and CTLA-4 needs to be clarified. In addition, p53 has also been reported to be a suppressor of T cell proliferation[23]. A lack of p53 inhibited activation-induced cell death and increased proliferation, the expression of stemness gene signature and antitumour activities[33,34]. Indeed, p53 expression in iT$_{SCM}$ cells was lower than in iT$_{CM}$ cells after TCR stimulation. Thus, lower p53 expression may allow iT$_{SCM}$ cells to evade apoptosis and cell cycle arrest, and acquire antitumour activity, unlike iT$_{CM}$ cells. Clarifying the mechanism of downregulation of p53 in iT$_{SCM}$ cells will provide a new strategy for breaking immune checkpoint mechanisms.

In conclusion, although the mechanism underlying the induction of iT$_{SCM}$ cells by Notch signalling remained to be clarified, iT$_{SCM}$ cells should also prove useful in long-term potent vaccination against not only cancer but also severe infectious diseases. Our findings will pave the way to the establishment of highly efficient antigen-specific T cell therapy applicable for various clinical settings.

## Methods

**Mice.** Six- to eight-weeks-old male C57BL/6 mice were purchased from Tokyo Laboratory Animals Science (Tokyo, Japan) and Japan SLC (Shizuoka, Japan). IFN-γ Venus reporter mice were generated by introducing the transgenic BAC gene, which carried the IRES-Venus cassette in the *Ifng* exon 4. IL-4 human CD2 reporter mice were generated by introducing the transgenic BAC gene. IL-17A eGFP reporter mice (No. 018472) were ordered from Jackson Labs (Bar Harbor, ME, USA). OT-I-TCR-transgenic[35], OT-II-TCR-transgenic[36], and *Rag2*$^{-/-}$ (ref. 37) mice were described previously. All mice were kept in specific pathogen-free facilities at Keio University. Animal experiments were performed in strict accordance with the recommendations in the Guidelines for Proper Conduct of the Animal Experiments of the Science Council of Japan. All experiments using mice were approved by the Animal Ethics Committee of Keio University and were performed according to the Animal Ethics Committee's guidelines.

**Antibodies and reagents.** Fluorophore-conjugated or biotinylated monoclonal anti-mouse CD3ε (145-2C11), anti-mouse CD4 (RM4-5), anti-mouse CD8α (53-6.7), anti-human/mouse CD11b (M1/70), anti-mouse CD11c (N418), anti-mouse CD19 (6D5), anti-human/mouse CD45R (RA3-6B2), anti-mouse CD49b (DX5), anti-mouse TER-119 (TER-119), anti-mouse TCRγδ (GL3), anti-mouse IFN-γ (XMG1.2), anti-mouse IL-4 (11B11), anti-mouse/Rat IL-17A (eBio17B7), anti-mouse/Rat Foxp3 (FJK-16s), anti-mouse/human CD44 (IM7), anti-mouse CD62L (MEL-14), anti-mouse CD45.1 (A20), anti-mouse CD45.2 (104), anti-mouse Notch1 (HMN1-12), anti-mouse Notch2 (HMN2-35), anti-mouse Notch3 (HMN3-133), anti-mouse KLRG1 (2F1), anti-mouse Ly-6C (HK1.4), anti-mouse CD127 (A7R34), anti-mouse CD25 (PC61.5), anti-mouse CD154 (MR1), anti-mouse/Rat CD278 (C398.4A), anti-mouse CD279 (J43), anti-mouse CD152 (UC10-4B9), anti-mouse CD197 (4B12), anti-mouse CD122 (TM-β1), anti-mouse Ly-6A/E (E13-161.7), anti-mouse Bcl-2 (BCL/10C4), anti-Human/Mouse phospho-AKT (S473) (SDRNR), anti-human CD2 (RPA-2.10), anti-human CD4 (RPA-T4), anti-human CD8α (HIT8a), anti-human CD45RA (HI100), anti-human CD45RO (UCHL1), anti-human CCR7 (G043H7), anti-human CD95 (DX2), anti-human CD154 (24-31), anti-human CD279 (PD-1) (EH12.2H7), anti-human CD152 (CTLA-4) (L3D10), and anti-human IFN-γ (B27) were purchased from eBioscience (San Diego, CA, USA) or Bio Legend (San Diego, CA, USA). APC-conjugated Ab specific for mouse CXCR3 (220803) and Recombinant Mouse DL1 Fc Chimera Protein were purchased from R&D systems (Minneapolis, MN, USA). PE-conjugated Ab specific for human CD8β (2ST8.5H7) was purchased from Beckman Coulter (Brea, CA, USA). Alexa Fluor 647-conjugated Ab specific for mouse phospho-Stat5 (Y694) (47/Stat5(pY694)) was purchased from BD Biosciences (Franklin Lakes, US-NJ, USA). Alexa Fluor

647-conjugated mouse p53 was purchased from Cell Signaling Technology (Danvers, MA 01923, USA). HLA-A2 Mart-1 (Alexa Fluor 647-labelled HLA-A-02:01/ELAGIGILTV) and HLA-A2 PSA (Alexa Fluor 647-labelled HLA-A-02:01/FLTPKKLQCV) tetramers were obtained from the NIH tetramer core facility at Emory University. γ-secretase inhibitor DAPT was purchased from Cayman Chemical (Ann Arbor, MN, USA). ABCG2 inhibitor FTC was purchased from Bioaustralis (Smithfield, NSW, Australia). FK506 was purchased from Sigma-Aldrich (St. Louis, MO, USA). OVA peptides (OVA$_{257-264}$ and OVA$_{323-339}$) were purchased from Synpeptide (Shanghai, China). A CellTrace™ CFSE Cell Proliferation Kit (#C34554), Propidium iodide (#P3566), Vybrant DyeCycle Violet Stain (DCV, # V35003), and LIVE/DEAD Viability/Cytotoxicity kits for mammalian cells containing calcein AM and ethidium homodimer-1 (#L3224) were purchased from Thermo Fisher Scientific (Waltham, MA, USA). An AnnexinV-FITC Apoptosis kit (#K101-25) was purchased from BioVision (Milpitas, CA, USA).

**Cell lines.** OP9 cells were a kind gift from Dr. Hiroshi Kawamoto (Kyoto University). Control OP9 feeder cells, mouse Delta-like1 expressing-OP9 cells (OP9-DL1), and human Delta-like 1 expressing-OP9 cells (OP9-hDL1) were cultured in alpha MEM (Thermo Fisher Scientific) supplemented with 20% FBS and 1% penicillin/streptomycin. E.G7-OVA cells were purchased from ATCC (Manassas, VA USA). E.G7-OVA cells were cultured in complete RPMI 1640 (Thermo Fisher Scientific) supplemented with 10% FBS, 1% penicillin/strepto-mycin, and 400 μg ml$^{-1}$ G418 (Nacalai Tesque, Kyoto, Japan). Lymphoblastoid cell lines (LCLs) were established from peripheral blood mononuclear cells (PBMCs) provided from three healthy donors, as described in previous studies[25,26]. PBMCs were cultured in complete RPMI medium with 10% FBS, Epstein–Barr virus (EBV) solution, and 20 nM FK506 for more than one month.

**Mouse primary T cell and CD11c$^+$ DC isolation.** Mouse CD4$^+$ T cells were isolated from spleen and lymph nodes using a mouse CD4$^+$ T cell isolation kit (#130-104-454) and AutoMACS Pro (Miltenyi Biotec, Bergisch Gladbach, Germany). Isolated CD4$^+$ T cells were stained with fluorophore-conjugated anti-CD4/CD8α, and CD44, CD62L Abs and naive CD4$^+$ T cells were purified by gating on CD4$^+$/CD8α$^+$CD25$^-$CD44$^{lo}$CD62L$^{hi}$ cells using a cell sorter.

Mouse CD8$^+$ T cells were isolated from spleen and lymph nodes by depletion of non-CD8α$^+$ T cells using biotinylated Abs against mouse CD4, CD19, B220, CD11b, CD11c, CD49b, TER-119 and TCRγδ in combination with streptavidin MicroBeads using AutoMACS Pro (Miltenyi Biotec). Isolated CD8$^+$ T cells were stained with fluorophore-conjugated anti-CD8α, and CD44, CD62L Abs, and naive CD8$^+$ T cells were purified by gating on CD8α$^+$CD44$^{lo}$CD62L$^{hi}$ using a cell sorter. Splenic CD11c$^+$ dendritic cells (DCs) were positively selected using CD11c MicroBeads, mouse (#130-108-338, Miltenyi Biotec) from splenocytes of 6- to 8-week-old wild-type mice.

**Activation of primary mouse CD4$^+$ and CD8$^+$ T cells *in vitro*.** Wild-type mouse naive CD4$^+$ T cells were stimulated with soluble anti-CD28 Ab (57.31; 2 μg ml$^{-1}$) and plate-coated anti-CD3ε Ab (145-2C11; 5 μg ml$^{-1}$) under each set of Th differentiation conditions. The conditions in this study were as follows: Th1 conditions: IL-12 (PeproTech, Rocky Hill, NJ; 10 ng ml$^{-1}$) and anti-IL-4 Ab (11B11; 2 μg ml$^{-1}$); Th2 conditions: IL-4 (PeproTech; 20 ng ml$^{-1}$) and anti-IFN-γ Ab (R4-6A2; 2 μg ml$^{-1}$); Th17 (TGF-β + IL-6) conditions: TGF-β (Bio Legend; 2 ng ml$^{-1}$), IL-6 (PeproTech; 30 ng ml$^{-1}$), anti-IFN-γ Ab (R4-6A2; 2 μg ml$^{-1}$), anti-IL-4 Ab (11B11, 2 μg ml$^{-1}$), and anti-IL-2 Ab (JES6-1A12, 2 μg ml$^{-1}$); Th17 (IL-1β + IL-6 + IL-23) conditions: IL-1β (PeproTech; 20 ng ml$^{-1}$), IL-6 (PeproTech; 30 ng ml$^{-1}$), IL-23 (PeproTech; 20 ng ml$^{-1}$), anti-IFN-γ Ab (R4-6A2; 2 μg ml$^{-1}$), anti-IL-4 Ab (11B11, 2 μg ml$^{-1}$), and anti-IL-2 Ab (JES6-1A12, 2 μg ml$^{-1}$); and Treg conditions: TGF-β (Bio Legend; 2 ng ml$^{-1}$), IL-2 (PeproTech; 20 ng ml$^{-1}$), anti-IFN-γ Ab (2 μg ml$^{-1}$) and anti-IL-4 Ab (11B11, 2 μg ml$^{-1}$).

*Rag2*$^{-/-}$ OT-II and OT-II naive CD4$^+$ T cells were primed with 10 μg ml$^{-1}$ OVA$_{323-339}$ peptide pulsed splenic CD11c$^+$ DCs under Th1 conditions, and OT-I naive CD8$^+$ T cells were primed with 10 μg ml$^{-1}$ OVA$_{257-264}$ pulsed splenic CD11c$^+$ DCs with 20 ng ml$^{-1}$ IL-2.

All cultures were grown in RPMI 1640 (Invitrogen) supplemented with 10% FBS, 1% penicillin/streptomycin, 100 nM nonessential amino acids, 2 mM glutamine and 0.05 mM β-mercaptoethanol.

**Human T cell culture.** Human PBMCs were prepared by specific gravity centrifugal methods from peripheral blood. Peripheral blood was provided by three healthy donors who were EBV-seropositive (VCA-IgG [ + ], EBNA [ + ]). We prepared CD8$^+$ T cells from PBMCs with depletion of non-CD8α$^+$ T cells using a human CD8$^+$ T cell isolation kit (#130-096-495, Miltenyi Biotec). To induce EBV-specific CD8$^+$ T cells, CD8$^+$ T cells were labelled with CFSE and were subsequently cocultured with 40 Gy-irradiated autologous LCLs for seven days. To induce Flu-specific CD8$^+$ T cells, we labelled PBMC with CFSE and subsequently cultured them in the presence of heat-killed influenza A virus (PR8 line). We prepared CD4$^+$ T cells from PBMCs with depletion of non-CD4$^+$ T cells through human CD4$^+$ T cell isolation kit (#130-096-533, Miltenyi Biotec).

To induce EBV-specific CD4$^+$ T cells, CD4$^+$ T cells were labelled with CFSE and were subsequently cocultured with 40 Gy-irradiated autologous LCLs for 7 days. This study was approved by the Ethics Committee of Keio University.

**OP9-DL1 or OP9-hDL1 cell coculture system.** Mouse and human T cells were activated using the methods mentioned above. To activate Notch signalling, activated T cells were cocultured with Notch ligand-expressing OP9 feeder cells, OP9-DL1 cells for mouse T cells and OP9-hDL1 cells for human T cells. We cultured mouse T cells and OP9-DL1 cells with mouse IL-7 (PeproTech; 10 ng ml$^{-1}$), an anti-mouse IFN-γ antibody (R4-6A2; 2 μg ml$^{-1}$), in alpha MEM for 11–12 days. Human T cells and OP9-hDL1 cells were cocultured with human IL-7 (PeproTech; 10 ng ml$^{-1}$), an anti-human IFN-γ antibody (Bio Legend, B27; 2 μg ml$^{-1}$), in alpha MEM for 11 days.

**Flow cytometry and cell sorting.** Antibody dilution factors are 1:400 for antibodies purchased from BioLegend and eBioscience, and 1:100 for antibodies purchased from R&D systems, Beckman Coulter, BD Biosciences and Cell Signaling Technology and for HLA-A2 tetramers. For IFN-γ, IL-4, IL-17A and Foxp3 intracellular staining, cells were stimulated for 6 h in a complete medium with PMA (50 ng ml$^{-1}$) and ionomycin (500 ng ml$^{-1}$; both from Sigma-Aldrich) in the presence of brefeldin A (eBioscience). Surface staining was then performed in the presence of Fc-blocking antibodies (2.4G2), followed by intracellular staining for anti-IFN-γ, IL-4, IL-17A and Foxp3 antibodies with Foxp3/transcription factor fixation/permeabilization concentrate and diluent (#00-5521-00, eBioscience), according to the manufacturer's instructions. For pStat5 and pAkt staining, cells were fixed and permeabilized with 4% paraformaldehyde and ice-cold 90% methanol and incubated with antibodies for 1 h at room temperature. For Bcl-2 and p53 staining, cells were fixed and permeabilized with Foxp3/transcription factor fixation/permeabilization concentrate and diluent (eBioscience), according to the manufacturer's instructions. We performed flow cytometry acquisition on a FACS Canto II cytometer (BD Biosciences, San Jose, CA, USA) and analysed the data using FlowJo software (Tree Star, Ashland, OR, USA). We sorted mouse and human T cells with a FACS Aria II, FACS Aria III cell sorter (BD Biosciences) and an SH800 cell sorter (Sony, Tokyo, Japan).

**In vitro and in vivo stimulation of iT$_{SCM}$ cells.** For in vitro proliferation assay, naive CD4$^+$ T cells, T$_{EM}$, T$_{CM}$ cells, iT$_{CM}$ and iT$_{SCM}$ cells generated from in vitro activated Rag2$^{-/-}$ OT-II CD4$^+$ T cells were CFSE labelled ($5 \times 10^4$), then stimulated with $1.25 \times 10^4$ OVA-DCs for indicated periods. For in vivo proliferation assay, each T cell subset was obtained from CD45.1$^+$ CD4$^+$ OT-II mice. Naive T cells, T$_{EM}$ cells, T$_{CM}$ cells, CD62L$^+$ cells induced by OP9 coculture, iT$_{CM}$, and iT$_{SCM}$ cells ($2 \times 10^5$) were transferred into CD45.2$^+$ mice, followed by OVA/IFA immunization. T cells were analysed on day 3 or on day 6.

**Homoeostatic short and long-term survival assay.** All T cells were prepared from Rag2$^{-/-}$ OT-II mice. For short term proliferation, we injected CFSE-labelled $5 \times 10^5$ CD45.2$^+$ T cells into the sublethally irradiated recipient CD45.1$^+$ congenic mice, then 20 days later, T cells were isolated from spleen and LNs and analysed. For long-term survival assay in vivo, CFSE-labelled $5 \times 10^4$ CD45.2$^+$ T cells were adoptively transferred into sublethally irradiated CD45.1$^+$ congenic mice. After 150 days, numbers of the CD45.2$^+$ T cells in the spleen, the peripheral lymph nodes (pLN), and the mesenteric LN (mLN) were determined. For serial transfer experiments, CFSE-labelled $1 \times 10^6$ CD45.2$^+$ T cells were injected into the sublethally irradiated recipient CD45.1$^+$ congenic mice (first transfer), then 20 days later, T cells were isolated from spleen and LNs and CFSE$^{lo}$CD44$^{lo}$CD62L$^{hi}$ fraction for iT$_{SCM}$ cells or naive T cells, or CFSE$^{lo}$CD44$^{hi}$CD62L$^{hi}$ fraction for iT$_{CM}$ cells were sorted by FACS. These cells were then re-labelled with CFSE and transferred into the sublethally irradiated recipient mice (second transfer). Then 20 days later, the same procedure was performed as third transfer. Mice of third transfer were analysed on day 10 after transfer.

**Adoptive transfer and tumour models.** We intravenously injected isolated T cells into wild-type mice via the tail vein. Recipient mice were sublethally irradiated at 4.5 Gy before adoptive transfer. CD45.2$^+$ T cells were transferred into CD45.1$^+$ congenic mice, while CD45.1$^+$ congenically labelled T cells were transferred into CD45.2$^+$ WT mice. For serial transfer experiments, CFSE-labelled CD4$^+$ T cells were transferred into mice for the primary transfer. Twenty days later, CFSE$^{lo}$CD4$^+$ T cells were sorted from the spleen and LNs of the primarily or secondarily transferred mice, and then transferred into the next-step mice.

The tumour transplantation and therapeutic model was performed according to the procedure reported by Gattinoni et al.[12]. E.G7-OVA ($1 \times 10^7$) cells were intradermally injected eight days before T cell transfer to establish therapeutic models. T cells ($3 \times 10^5$) were adoptively transferred into tumour-bearing mice in conjunction with emulsified incomplete Freund's adjuvant (IFA) containing OVA (250 μg per mouse). Mice were intraperitoneally injected with 500 ng IL-2 twice, 24 and 48 h after OVA immunization. Tumour sizes were measured with a vernier caliper every 2–3 days, and tumour volume was calculated according to the

following formula: volume = 0.5 × length × width$^2$. We set a humane endpoint in the tumour model; mice were sacrificed when the tumour volume reached 2000 mm$^3$, which has been approved by the Animal Ethics Committee of Keio University.

**Side population analysis.** Side population analysis was performed using a violet-excited DNA dye, Vybrant DyeCycle Violet Stain (DCV), as described previously[35]. Briefly, cells were incubated with a violet-excited dye, Vybrant DCV Stain, for 30 min at 37 °C in the presence or absence of a ABCG2 inhibitor, FTC (10 μM) in RPMI medium with 10% FBS. To inhibit DCV efflux in T cells, positive control samples were cultured with FTC.

**CTL assay.** LIVE/DEAD Viability/Cytotoxicity kits for mammalian cells containing calcein AM and ethidium homodimer-1(#L3224) were obtained from Thermo Fisher Scientific. Isolated EBV-specific or Flu-specific CD8$^+$ T cells were cocultured with autologous LCL in complete RPMI medium at 37 °C for 3 h. Collected cell suspension was stained with calcein AM and ethidium homodimer-1 (EthD-1) after incubation. Dead LCL cells were detected as CD8α$^-$ calcein AM$^-$EthD-1$^+$ cells by flow cytometry analysis.

**Microarray analysis.** For mouse microarray analysis, total RNA was isolated using a ReliaPrep RNA Cell Miniprep System (#Z6012, Promega, Madison, WI, USA). Microarray analysis was performed by TakaraBio using SurePrint G3 Mouse Gene Expression 8 × 60K (Agilent, Santa Clara, CA, USA). The expression values were calculated using Feature extraction software (Agilent Technologies). The values were normalized by adjusting each expression data to a 75th percentile baseline, following log2 transformation. (The normalization step was performed with an original implementation of the 75th percentile normalization algorithm written in C.) Further statistical analyses were performed using the R statistical language and environment. The graphics generated and tools of analysis used are available at http://discover.nci.nih.gov. The data for mouse samples were deposited in GEO database (GSE 92381).

For the microarray analysis of human samples, all data, except the calculation of FKPM values of NexSeq data, were analysed by statistical computing, using the R language. The expression levels and expression barcodes of Affymetrix Human Gene 1.0 ST Array data (GSE23321, GSE68003 and GSE93211) were calculated from.CEL files using the 'frma' function of the frma package. The SRA files from Illumina NextSeq 500 (GSE80306) were converted to fastq files by a fastq-dump of the SRA toolkit and then mapped to an Ensemble GRCh37/hg19 genome annotation using STAR. FPKM values of uniquely mapped reads were calculated by Cufflinks. Before combining the ST array and NextSeq data, low or no expressed genes were removed by the following method: In the ST array, we selected 2496 genes which are estimated to be 'Gene Expression Barcode = 1' by barcode function. In NextSeq, 8168 genes were selected by FKPM values > 2. The expression levels of 1771 genes that overlapped in the ST array and NextSeq data were converted to log2, and the batch effects between the data set and platform were removed using the 'calcResiduals' function of the pcbcStats package.

Principal component analysis were performed by 'prcomp' function of stats package, respectively and visualized by ggplot2 package. Clustering using cosine distance and the Ward's linkage method were performed using the 'Heatmap' function of the ComplexHeatmap package. Statistical tests between cell types were performed using the eBays function of the limma package. Then, 1600 genes among 7 cell types were selected by an FDR adjusted P value < 0.01. (GO component) pathway analysis of the selected genes was performed by Fisher's exact test using the 'fisher test' function of the stats package. The top 30 molecular concepts with P values < 0.0001 were visualized using the ComplexHeatmap package.

**Quantitative real-time PCR.** Total RNA was extracted using RNAiso Plus (#Z6012, Takara Bio, Shiga, Japan) or ReliaPrep RNA Miniprep Systems (#4368814, Promega) and subjected to reverse transcription using a High Capacity complementary DNA (cDNA) Synthesis Kit (Thermo Fisher Scientific). PCR analysis was performed using an iCycler iQ multicolour real-time PCR detection system (Bio-Rad, Hercules, CA, USA) and SsoFast EvaGreen Supermix (Bio-Rad). All primer sets yielded a single product of the correct size. Relative expression levels were normalized to Hprt. Specific primers are described in Supplementary Table 1.

**Plasmid construction and retroviral transduction.** Mouse Notch1 and Notch2 intracellular domain (ICD) were PCR-amplified from the mouse cDNA library and subcloned into eMIThyR vectors containing the MCS-IRES-Thy1.1 sequence. Retroviral transduction was performed, as described in ref. 36. Briefly, naive T cells were plated and subjected to the Th1-differentiation conditions described above starting on day 0. On Day 2, fresh retroviral supernatants were added, and the cells were centrifuged at 2,500 r.p.m. for 2 h at 35 °C. After spin infection, the cells were cultured in the appropriate Th cell differentiation media and collected on day 4 for OP9 coculturing.

**ELISA.** For analysis of IL-2 in a cultured medium, the supernatants were analysed with a Mouse IL-2 ELISA Ready-SET-Go! (#88-7024-77, eBioscience), according to the manufacturer's protocol.

**Cell cycle and apoptosis detection assays.** To detect cell cycle, cells were stimulated with anti-CD3 antibody ($5 \mu g \, ml^{-1}$) and anti-CD28 antibody ($2 \mu g \, ml^{-1}$) for zero, one, two and three days. Collected cells were suspended in ice-cold 90% methanol and incubated at $-30\,^{\circ}C$ overnight for fixation and permeabilization. Fixed and permeabilized cells were re-suspended in propidium iodide (PI)-containing buffer and DNA content at indicated time points was analysed by flow cytometry. Cells in the G2/M phase, S phase and G0/G1 phase were detected as $PI^{high}$, $PI^{int}$ and $PI^{low}$-stained cells, respectively.

To detect apoptosis, cells were stimulated with anti-CD3 antibody ($5 \mu g \, ml^{-1}$) and anti-CD28 antibody ($2 \mu g \, ml^{-1}$) for 60 h. Apoptotic cells were stained by AnnexinV-FITC Apoptosis kit (BioVision) and analysed by flow cytometry.

**Study approval.** This study was approved by the Institutional Review Board of Keio University School of Medicine (Approval number: 20120039), and conducted in compliance with the Declaration of Helsinki. Written informed consent was obtained from all individuals.

**Statistics.** Statistical analysis was performed using the two-tailed unpaired Student's $t$-test, Mann-Whitney test, one-way ANOVA, two-way ANOVA and Kaplan–Meier method, using GraphPad Prism version 6.05 software (GraphPad Software, CA, USA). One-way ANOVA and two-way ANOVA were used for multiple comparisons. The variance among the groups was estimated using the F-test, and $P$ values $< 0.05$ were considered statistically significant. All data are presented as the mean ± s.e.m. Mice were randomly assigned to experimental groups. The investigators were not blinded to allocation during experiments and outcome assessment.

**Data availability.** The data that support the findings of this study are available within supplementary information files or from the corresponding author on reasonable request. Microarray data that support the findings of this study have been deposited in GEO with the primary accession codes GSE92381 (mouse) and GSE93211 (human).

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

## Acknowledgements

We thank Mayako Asakawa (Keio University) for preparing the virus stock; Minako Ito, Noriko Shiino, Yoshiko Noguchi, Yasuko Hirata and Yukiko Tokifuji (Keio University) for their technical assistance; Yuki Ushijima (Keio University) for manuscript preparation; Kazuhiko Takahara (Kyoto University) and Ryusuke Muro (The University of Tokyo) for providing materials; Taeka Hayakawa, Hajime Kamijuku, Tomonori Yaguchi and Yutaka Kawakami (Keio University) for discussions; and the NIH Tetramer Facility for reagent preparation. OP9 cells and OP9-DL1 cells were kindly provided by Hiroshi Kawamoto (Kyoto University). This work was supported by special Grants-in-Aid from the Ministry of Education, Culture, Sports, Science and Technology of Japan (No. 25221305), Advanced Research & Development Programs for Medical Innovation (AMED-CREST), the Takeda Science Foundation, the Uehara Memorial Foundation, the Mochida Memorial Foundation for Medical and Pharmaceutical Research, the Kanae

Foundation, the SENSHIN Medical Research Foundation and the Keio Gijuku Academic Developmental Funds.

## Author contributions

T.K. and R.M. performed experiments; T.K., M.K. and Y.O. analysed data; Y.K., T.S., S.C., K.K., M.K., T.S., H.N. and T.M. edited the manuscript; R.M. and A.Y. supervised the research; and T.K., R.M. and A.Y. wrote the manuscript.

## Additional information

**Competing interests:** The authors declare no competing financial interests.

