## [Peer Review File · Nature Communications]

Reviewers' comments:

Reviewer #1 (Remarks to the Author):

In this interesting article, Kondo et al demonstrate that Notch signaling can induce Tscm-like fate in activated T cells. This work extends prior advances from Gattinoni et al may be of importance for tumor immunotherapy. Overall, this work is of importance for the field, although there remain several issues to be clarified. These are listed below.

Major comments:

- In the course of their experiments, the authors generate two distinct phenotypes of cells by culturing activated cells with Notch ligands – naïve-like iTscm and memory-like iTcm. The authors then perform a long list of incisive experiments with these cells, but the key issue is their relationship to previously discovered human subsets, Tscm of Gattinoni et al. and Tmnp cells of Pulko et al. The authors did not resolve this issue, rather, they have named their cells something in between. So their iTscm cells have the phenotype more similar to Tmnp than to Tscm cells in mice but perhaps closer to Tscm than Tmnp in humans. The authors should make an effort to resolve this issue rather than introduce additional, potentially confusing, nomenclature into the field.
- In line with the above, human iTscm cells induced by Notch signaling really should not be called naïve-like – expression of CD95 clearly phenotypically classifies them as memory, as was the case in the original Tscm population described by Gattinoni.
 - When evaluating antitumor effects, it would be of importance to compare iTscm cells induced by Notch signals from activated cells to the same type of cells induced from naïve cells by Wnt signaling. This would help guide future immunotherapy efforts.
 - It is unclear what components of the culture system are essential for generation of iTscm cells – IL-7 and/or anti-IFN γ treatments should be analyzed individually in the presence of Notch signaling to that effect.
 - Not clear what “side populations” the authors refer to on p. 6 , second paragraph, last sentence.
 - Proliferative ability of human Notch-signaling induced iTscm was evaluated by IL-7. IL-15-mediated proliferation should be assessed too, as it will be informative relative to the differentiation status of these cells.

MINOR Comments

Overall, the paper would benefit from editing by a native English speaker. There are several imprecise and/or confusing formulations whose editing would improve the overall clarity of the manuscript. A few of them are listed below, but that list is not exhaustive:

p5, bottom paragraph, first sentence – “deprived” is probably not the best word here. “Suppressed expression....” would be one better choice.

p6. , second paragraph, third sentence – formulation “ ...suggesting that CD4+iTSCMcells were likely memorizethe phenotypesof the original T cell subset”. It should be replaced with something like “suggesting that CD4+ iTscn are likely imprinted with cytokine-producing fate of the original....”

P9, third sentence from top – last word “done” should be replaced by “present”.

Reviewer #2 (Remarks to the Author):

This manuscript by Kondo et al. describes a discovery that could be very exciting and important. Recent years have brought tremendous excitement about the ability of adoptively transferred T cells to help eradicate cancers. One of the approaches showing promise is based on infusion of in vitro expanded T cells isolated from the patient’s own tumors. Many tumors contain T cells that recognize tumor antigens. Using T cells from tumors for therapy obviates the need to identify tumor antigens and/or introduction of T cell receptors. One of the hurdles is that with currently used procedures transferred cells tend to be of a short lived variety. While such cells may be

proficient at killing tumors initially, eventually their activity will subside. To prevent outgrowth of tumors that survived initial eradication by these T cells, it might be useful if transferred T cells had self renewal capacity. It has been reported that memory T cells with such stem cell like properties exist, although the phenotype of these cells is hotly debated. The best characterized memory T cell subsets are the so called effector memory T cells (Tem), which possess immediate effector capacity but limited ability to expand, and central memory T cells (Tcm), which have greater ability to expand and can generate both Tcm and Tem progeny. This latter property is reminiscent of stem cells and indeed it has been shown that these cells can maintain themselves in serial transplantation experiments. However, others have argued that true stem cell-like memory T cells exist, distinct from classical Tcm. Whichever of these options is correct, it would be useful if we could convert activated T cells (such as those obtained from tumors) into cells with stem cell properties.

Kondo et al. now report the discovery of a method that may allow such conversion. The method is deceptively simple and involved nothing more than activation of the Notch signaling pathway in activated T cells. Stimulation of Notch in activated T cells, obtained from various activation regimens, led a small proportion of these cells to acquire a phenotype reminiscent of naïve T cells (CD62L+CD44low). Further experiments indicated that these cells possess greater expansion and self renewal capacity than Tcm from the same cultures (which they refer to as induced Tcm or iTCM). Importantly, these induced stem cell T cell memory cells (iTSCM) were more effective at eradicating a model tumor in vivo. Finally, it is shown that cells with a similar phenotype can be obtained from activated human T cells.

Major criticism

For the most part, I believe that the experiments convincingly support the author's claims that iTSCM are superior to iTCM. I am surprised, however, that the latter cells perform very poorly in all the assays, and this makes me wonder about the correct interpretation of the results. Are the iTSCM cells particularly good or the iTCM particularly bad? In other words: does coculture of activated T cells diminish the abilities of the cells that emerge with a Tcm phenotype or does it promote these qualities in the iTSCM? In most of the experiments, there is no comparison of the iTSCM with bona fide Tcm obtained straight ex vivo or even with those cells that emerge from the control cultures. Without these references, I find myself reserved to support publication.

Minor points

1. It would be good if the authors could provide a figure showing the range and reproducibility of iTSCM induction at different time points. In Figure 1b, essentially no such cells are found after 4 days of coculture, whereas the data in Supplementary Figure 3 are all derived from 4 day cultures. If people want to reproduce these data, it would be important for them to know the expected variation.
2. The gate for the TCM in the OP9 samples is set much higher than in the OP9-Dll1 sample in for Figure 1b. If the iTSCM gate from the OP9-Dll1 sample (ranging till about 102) would be used in the OP9 samples, one would have to conclude that cells with a similar phenotype are present also in these cultures. Although I agree that the CD44/CD62L profile in the OP9 cultures looks different from that in the OP9-Dll1 samples, changing the gates between samples like the authors have done seems a bit arbitrary.
3. If you put the gates the same way in both cultures, how do the iTSCM from the OP9-Dll1 cultures compare in terms of phenotype and function to the few cells with cells that fall within the same gate in the OP9 control cultures? I would not ask to redo all the experiments with these cells, but at least phenotype and in vitro expansion capacity should be compared in my view.
4. Along similar lines, why is no OP9 control shown in Figure 1f and g?
5. I could not understand how the experiment was done in Figure 3a, so I cannot interpret it. I am assuming this figure is supposed to show the cell number obtained after activating iTSCM with OVA-DCs, but that is not clear from the legend. This should be explained more clearly.
6. What is the expansion in Fig 3a?
7. It is not clear to me whether the result in Figure 3b reflects superior expansion of the injected iTSCM cells or inferior expansion or survival of the other cell types. So few of the other cell types

are recovered that I wonder whether this result truly reflects on the relative expansion capacities of these cell types. How many cells were injected in Fig 3b?

8. How many cells were recovered in the homeostatic expansion experiments (figure 3c)?

9. Again, the set up of the experiment in Figure 3e is not clear to me. How many cells were transferred in each of the rounds? Was the same number of donor CD4 T cells transferred between groups in each round of transfer?

10. The representation of the result in Figure 5b does not allow me to assess how efficient the conversion into human iTSCM is, because the CD45RA/RO profiles shown have been pregated on CCR7+CD95+ cells. What proportion of the cultures is CCR7+CD95+?

11. What is the fold expansion in Figure 5d?

12. A value is missing for the Dltx1 mRNA graph in Supplementary figure 2d. Was this undetectable? What was the lower limit of detection? Are these data from a single experiment?

13. The number of cells recovered in Supplementary Figure 6c is very low (2.5% of input). What is the evidence that the cells expanded?

Point-by-point response to Reviewers' Comments:

Reviewer #1

Major comments:

•In the course of their experiments, the authors generate two distinct phenotypes of cells by culturing activated cells with Notch ligands – naïve-like iTscm and memory-like iTcm. The authors then perform a long list of incisive experiments with these cells, but the key issue is their relationship to previously discovered human subsets, Tscm of Gattinoni et al. and Tmnp cells of Pulko et al. The authors did not resolve this issue, rather, they have named their cells something in between. So their iTscm cells have the phenotype more similar to Tmnp than to Tscm cells in mice but perhaps closer to Tscm than Tmnp in humans. The authors should make an effort to resolve this issue rather than introduce additional, potentially confusing, nomenclature into the field.

We appreciate your important suggestion. Reviewer#1 pointed that “iT_{SCM} cells have the phenotype more similar to T_{MNP} than to T_{SCM} cells in mice”. We partly agreed with the reviewer; for example, CXCR3 expression levels were high in both human T_{MNP} and murine iT_{SCM} cells compared with murine T_{SCM} cells. To define the relationship among T_{MNP}, T_{SCM} and iT_{SCM} cells more precisely, we performed global gene expression analysis. Since T_{MNP} were reported in only in human, we compared DNA microarray data of our human CD8⁺ iT_{SCM} cells and gene expression profiles of CD8⁺ T cell subsets, deposited in the public database including T_{SCM} cells reported by Gattinoni et al. and T_{MNP} cells reported by Pulko et al. As shown in new **Supplementary Fig. 16** in the revised manuscript, Notch-induced iT_{SCM} cells were most close to “*in vitro* Wnt (TWS119)-induced T_{SCM}-cells” reported by Gattinoni et al. rather than with T_{MNP} cells. The cluster of Notch-induced iT_{SCM} cells and TWS119-induced T_{SCM} cells are distantly related to the cluster of naïve and T_{MNP} cells. Our data suggest that *in vitro* generation of T_{SCM}-like cells by either Notch or Wnt made the difference from *in vivo* T_{SCM} and T_{MNP} cells. There has not been described the method for *in vitro* generation of T_{MNP} cells. Therefore, we decided to continue to use the nomenclature “iT_{SCM}” for Notch-mediated T_{SCM}-like cells.

In vitro generation methods for T_{SCM}-like cells might be synthetic. However, we would like to emphasize that, even though such artificial pedigree, we believe *in vitro* generated T_{SCM}-like cells by Notch or Wnt signals are valuable for T-cell based immunotherapy.

•In line with the above, human iTscm cells induced by Notch signaling really should not be called naïve-like – expression of CD95 clearly phenotypically classifies them as memory, as was the case in the original Tscm population described by Gattinoni.

We agree with the reviewer. As pointed out by the reviewer, human iT_{SCM} cells had not phenotypically very close to naïve T cells. CD95 has been shown to be a marker of human T_{SCM} cells. Thus, we changed “naïve-like” to “T_{SCM}-like” in p12, 3rd paragraph, 4th sentence.

•When evaluating antitumor effects, it would be of importance to compare iT_{scm} cells induced by Notch signals from activated cells to the same type of cells induced from naïve cells by Wnt signaling. This would help guide future immunotherapy efforts.

We appreciate your comment. We compared antitumor effects between Wnt and Notch-induced iT_{SCM} cells and Wnt-induced T_{SCM} cells, as you suggested. As shown in new **Supplementary Fig. 13**, Notch-induced iT_{SCM} cells showed similar antitumor effects to those by Wnt-induced T_{SCM}-like cells. However, much larger number of iT_{SCM} cells was generated than TWS119-induced T_{SCM} cells from the same number of naïve OT-I CD8⁺ T cells. This is probably because of a strong growth inhibitory effect of TWS119. Therefore, our method using coculture with OP9-DL1 seems to be advantageous compared with Wnt-mediated method in terms of the cell number. These are included in the Result and Discussion sections.

•It is unclear what components of the culture system are essential for generation of iT_{scm} cells – IL-7 and/or anti-IFN γ treatments should be analyzed individually in the presence of Notch signaling to that effect.

We appreciate the reviewer's comment. We described the detailed necessity of cytokines and neutralizing antibodies. As shown in new **Supplementary Fig. 2e**, we found that IL-7 and anti-IFN- γ antibody are essential, since iT_{SCM} generation was severely reduced in the absence of IL-7 and anti-IFN- γ antibody (**Supplementary Fig. 2e**).

•Not clear what “side populations” the authors refer to on p. 6 , second paragraph, last sentence.

We appreciate your comment. Side population assay has been used to detect ABCG2 activity, which is one of the stem cell characters. We mentioned this in the text as follows:

Stem-like cells have been shown to express drug transporter including ATP-binding cassette sub-family G member 2 (ABCG2), which rapidly efflux lipophilic fluorescent dyes²⁰, thus exhibit so called “side population (SP)” fraction in flow cytometry. CD4⁺ iT_{SCM} cells showed more SP cells than CD4⁺ iT_{CM} T cells did (**Fig. 1e**), and this fraction was decreased by the

ABCG2 inhibitor Fumitremorgin C (FTC). These data suggest that CD4⁺ iT_{SCM} cells may have a characteristic feature related to stem cells.

•Proliferative ability of human Notch-signaling induced iTscm was evaluated by IL-7. IL-15-mediated proliferation should be assessed too, as it will be informative relative to the differentiation status of these cells.

We examined the effect of IL-15 on human iT_{SCM} generation. As shown in new **Supplementary Fig. 14d**, IL-15 markedly decreased the efficiency of iT_{SCM} generation regardless of the presence or absence of IL-7. The reason is not clear at present; however, these data indicate that IL-7 is essential for the generation of iT_{SCM} cells, while IL-15 prevents iT_{SCM} generation.

MINOR Comments

Overall, the paper would benefit from editing by a native English speaker. There are several imprecise and/or confusing formulations whose editing would improve the overall clarity of the manuscript. A few of them are listed below, but that list is not exhaustive:

p5, bottom paragraph, first sentence – “deprived” is probably not the best word here. “Suppressed expression....” would be one better choice.

p6. , second paragraph, third sentence – formulation “ ...suggesting that CD4+iTSCMcells were likely memorize the phenotypes of the original T cell subset”. It should be replaced with something like “suggesting that CD4+ iTscm are likely imprinted with cytokine-producing fate of the original....”

P9, third sentence from top – last word “done” should be replaced by “present”.

We greatly appreciate your comments. We corrected our mistakes as you pointed out. We also send our revised manuscript to English editing by a native speaker.

We hope now our manuscript is clearer.

Reviewer #2 (Remarks to the Author):

We appreciate favorable and constructive comments by the reviewer.

Major criticism

For the most part, I believe that the experiments convincingly support the author's claims that iTSCM are superior to iTCM. I am surprised, however, that the latter cells perform very poorly in all the assays, and this makes me wonder about the correct interpretation of the results. Are the iTSCM cells particularly good or the iTCM particularly bad?

We appreciate your valuable comments. We characterized iT_{CM} cells (induced by OP9-DL1) as well as CD62L⁺ cells (induced by OP9) more intensively and compared with iT_{SCM} cells. We also added data of *in vivo*-derived T_{CM} and T_{EM} cells which were induced by OVA/IFA immunization in OT-II mice and FACS sorted. As shown in new **Fig. 3a-c**, antigen-specific iT_{SCM} cells possessed much higher capacity not only to proliferate in response to antigen restimulation but also to survive for an extended time under homeostatic conditions than any other T cell subsets. The *in vitro* proliferation activity and the self-renewing ability of iT_{CM} cells was inferior to that of iT_{SCM} cells but close to *in vivo* T_{CM} cells (new **Fig. 3a,c**). As shown in new **Fig.3b** and **Supplementary Fig. 7d, e**, a larger number of iT_{CM} cells was recovered than that of *in vivo* T_{CM} cells and CD62L⁺ cells induced by OP9 coculture after transplantation and immunization. Furthermore, antitumor effects of iT_{CM} cells were equal to that of *in vivo* T_{CM} cells and CD62L⁺ cells (new **Supplementary Fig. 12c**). However, iT_{SCM} cells always showed highest antitumor activity.

In conclusion, we found that the functions of iT_{CM} cells are not so inferior to those of other T_{CM} phenotypic cells; however, iT_{SCM} cells had superior proliferative and self-renewing, and antitumor abilities than any other memory T cell subsets.

In other words: does coculture of activated T cells diminish the abilities of the cells that emerge with a Tcm phenotype or does it promote these qualities in the iTSCM?

We think that Notch signals induce iT_{SCM} characters because iT_{SCM} cells exhibited the superior ability than any other memory T cell subsets and iT_{CM} cells were similar to other T_{CM} cells. In addition, new **Supplementary Fig.16** shows that Notch-induced iT_{SCM} cells are most close to *in vitro* TWS119-induced T_{SCM}-cells and are distantly related to the cluster of other T cell subsets. Thus, we believe that the coculture with OP9-DL1 cells (i.e. Notch signaling) promotes the qualities in iT_{SCM} cells. Unfortunately, the mechanism of the development of two different population; iT_{SCM} and iT_{CM} during OP9-DL1 coculture is not known at present.

In most of the experiments, there is no comparison of the iTSCM with bona fide Tcm obtained straight ex vivo or even with those cells that emerge from the control cultures. Without these references, I find myself reserved to support publication.

Your comment is very important to understand the unique characteristics of iT_{SCM} cells. As mentioned in the first question, we compared naive T cells, activated T cells, iT_{SCM} cells and iT_{CM} cells induced by OP9-DL1 coculture, CD62L⁺ cells induced by OP9 coculture, and *in vivo*-derived T_{CM} and T_{EM} cells. T_{CM} and T_{EM} cells were isolated by FACS from OT-II mice immunized with OVA/CFA. These data are shown in new **Fig.3 a-c**, new **Supplementary Fig. 7d,e** and **Supplementary Fig. 12c**. We concluded that iT_{SCM} cells had superior proliferative and self-renewing, and antitumor abilities than any other memory and naive T cell subsets.

Minor points

1. It would be good if the authors could provide a figure showing the range and reproducibility of iTSCM induction at different time points.

Thank you for your important suggestion. We amended and added the figures for the iT_{SCM} generation process. As shown in new **Fig.1b, c** and **Fig.2a** (mouse) and new **Supplementary Fig. 14b** (human), we showed gates and the value of fraction (%) at all time points. The experimental number of iT_{SCM} generation, mean and s.e.m were described in figure legend. Time courses of generation of iT_{SCM} cells are shown in new **Fig.1c** (mouse) and **Supplementary Fig. 14b** (human).

In Figure 1b, essentially no such cells are found after 4 days of coculture, whereas the data in Supplementary Figure 3 are all derived from 4 day cultures. If people want to reproduce these data, it would be important for them to know the expected variation.

We are sorry for making you confusion. Generation of CD44^{lo}CD62L^{hi} iT_{SCM} cells from OT-I or OT-II activated T cells takes 11-12 days after co-culture with OP9-DL1, while those from WT T cells required only 4 days (in **Supplementary Fig. 3**). We mentioned this in the Result section (p5 lines 14-15). We suspect that this may be due to different priming conditions by OVA-DCs and anti-CD3ε antibody, however, the precise reason of this difference is not clear at present.

2. The gate for the TCM in the OP9 samples is set much higher than in the OP9-Dll1 sample in for Figure 1b. If the iTSCM gate from the OP9-Dll1 sample (ranging till about 102) would be used in the OP9 samples, one would have to conclude that cells with a similar phenotype are present also in these cultures. Although I agree that the CD44/CD62L profile in the OP9 cultures looks different from that in the OP9-Dll1 samples, changing the gates between samples like the authors have done seems a bit arbitrary.

We appreciate your valuable comments. We addressed this question by comparing CD44^{hi} and CD44^{lo} factions in CD62L⁺ cells generated after coculture with OP9 cells. The gate of later fraction is close to that of iT_{SCM} induced by OP9-DL1 coculture. We compared three types of cells including whole CD62L⁺ cells induced by OP9 coculture with iT_{CM}/iT_{SCM} cells induced by OP9-DL1 coculture.

First, we compared surface marker expressions between CD44^{hi} and CD44^{lo} factions in CD62L⁺ cells induced by OP9 coculture. As shown in new **Supplementary Fig. 4**, there were no differences in the surface marker expressions between CD44^{lo} cells and CD44^{hi} cells. Furthermore, as shown in new **Supplementary Fig. 7b**, isolated CD44^{lo} cells as well as CD44^{hi} cells were similarly less proliferative after re-stimulation with antigen-loaded DCs compared with iT_{SCM} cells.

Then we compared phenotypical features among whole CD62L⁺ cells and iT_{CM}/iT_{SCM} cells. As shown in new **Fig. 1d,e**, CD62L⁺ cells expressed high levels of CTLA4 and low levels of CCR7. Thus CD62L⁺ cells were phenotypically distinct from iT_{CM} cells, indicating that CD62L⁺ cells induced by OP9 coculture are not identical to iT_{CM} cells induced by OP9-DL1. This suggests that iT_{CM} cells still received some Notch signaling.

In conclusion, CD44^{hi} and CD44^{lo} cells induced by OP9 coculture are similar but apparently different from iT_{SCM} cells induced by OP9-DL1 coculture. As we did not find any differences between CD44^{hi} and CD44^{lo} faction in surface markers and proliferation, we used whole CD62⁺ cells for further characterization.

3. If you put the gates the same way in both cultures, how do the iTSCM from the OP9-Dll1 cultures compare in terms of phenotype and function to the few cells with cells that fall within the same gate in the OP9 control cultures? I would not ask to redo all the experiments with these cells, but at least phenotype and in vitro expansion capacity should be compared in my view.

We appreciate this important comment. To address your questions, we performed additional experiments as your suggestions. We have isolated CD44^{lo} cells after OP9 coculture by similar gating to that of iT_{SCM} cells. It was impossible to analyze cells expressing very low levels of CD44 due to extremely small number of cells. As mentioned above, surface marker

expressions in CD44^{lo} cells (similar gating to that of iT_{SCM} cells) were similar to those in CD44^{hi}, but different from iT_{CM} and iT_{SCM} cells. CD44^{lo} cells as well as CD44^{hi} cells were less proliferative after re-stimulation with antigen-loaded DCs compared with iT_{SCM} cells (**Supplementary Fig. 7b**). In addition, as shown in new **Fig.3c**, the growth of T cells on OP9 feeder was much poor compared with that on OP9-DL1. Therefore, Notch signaling induced by OP9-DL1 seems to be necessary to generate iT_{SCM} cells, suggesting that CD62L⁺ cells are qualitatively different from iT_{SCM} and iT_{CM} cells.

4. Along similar lines, why is no OP9 control shown in Figure 1f and g?

Thank you for your comment. We added the OP9 control (CD62L⁺ T cells) in new **Fig. 1i** as you suggested. Similar to **Fig.1b**, CD44^{lo}CD62L^{hi} iT_{SCM} fraction rarely appeared after coculture with OP9 feeder cells, while significant CD44^{lo}CD62L^{hi} iT_{SCM} cells developed after OP9-DL1 coculture. We have not performed microarray analysis for CD62L⁺ T cells induced by OP9 coculture since CD62L⁺ T cells are very different from iT_{SCM} cells as shown in the reply for your questions 2 and 3. In this experiment, we intended to show the difference between iT_{SCM} and iT_{CM}.

5. I could not understand how the experiment was done in Figure 3a, so I cannot interpret it. I am assuming this figure is supposed to show the cell number obtained after activating iTSCM with OVA-DCs, but that is not clear from the legend. This should be explained more clearly.

We are sorry about incomplete description about experimental conditions and for making you confused. Previous Fig.3a did not exactly represent proliferation capacity of iT_{CM} and iT_{SCM} cells after antigen-restimulation. Thus we replaced it with new **Fig3a**, which showed CFSE-dilution assay after *in vitro* restimulation with OVA-DCs. This data showed that iT_{SCM} cells responded most rapidly than any other T cell subsets including naive T, iT_{CM}, T_{EM} and T_{CM} cells. The final cell division of iT_{SCM} cells after 60 h stimulation was similar to that of naive T and iT_{CM} cells but still higher than *in vivo*-derived T_{EM} and T_{CM} cells. Previous **Fig.3a** (now moved to new **Supplementary Fig.7c**) experiment showed proliferation activity of iT_{CM} and iT_{SCM} cells during the secondary co-culture on OP9-DL1. These experiments demonstrated that iT_{SCM} cells, but not iT_{CM} cells, retained a high proliferation potential on OP9-DL1 feeder cells. Sorry about our miss-presentation.

6. What is the expansion in Fig 3a?

Sorry about our miss-presentation. As mentioned, we moved this figure to new **Supplementary Fig.7c**, and “expansion” was changed to actual “cell number”.

7. It is not clear to me whether the result in Figure 3b reflects superior expansion of the injected iTSCM cells or inferior expansion or survival of the other cell types. So few of the other cell types are recovered that I wonder whether this result truly reflects on the relative expansion capacities of these cell types. How many cells were injected in Fig 3b?

We appreciate your comment. When the cell number on day 3 (**Fig.3b**) and on day 6 (**Supplementary Fig.7d**) were compared, the number of all T cell subsets were increased. CFSE-dilution assay on day 6 (new **Supplementary Fig.7e**) also indicated cell division occurred in all cell types. Thus, we think that iT_{SCM} cells proliferated faster than other cell types and may be resistant to activation-induced cell death as shown *in vitro* (new **Supplementary Fig.10b** and **Supplementary Fig.11b**). We injected 2×10^5 cells into single mouse but please note that not all cells can be recovered from the spleen and LN. We obtained $3-4 \times 10^5$ iT_{SCM} cells in the spleen on day 6 indicating that iT_{SCM} cells were actually expanded *in vivo*.

8. How many cells were recovered in the homeostatic expansion experiments (figure 3c)?

We added the graph showing actual cell number of the recovered cells. As shown in new **Fig. 3c lower right panel**, approximately $3-10 \times 10^3$ naïve T and T_{CM} cells, $1-2 \times 10^3$ iT_{CM} cells and $17-21 \times 10^3$ cells (iT_{SCM}) were recovered.

9. Again, the set up of the experiment in Figure 3e is not clear to me. How many cells were transferred in each of the rounds? Was the same number of donor CD4 T cells transferred between groups in each round of transfer?

We are sorry about incomplete description about experimental conditions. We added detail in the figure legend and M&M section. We performed this experiment according to the procedure described by Gattinoni, L., *et al. Nature medicine* **15**, 808-813 (2009) ref. 13 (Supplementary Fig. 8).

First, we injected CFSE-labeled 1×10^6 cells into the recipient mice (1st transfer), then 20 days later, T cells were isolated from spleen and LNs and CFSE^{lo}CD44^{lo}CD62L^{hi} fraction for iT_{SCM} cells or naïve T cells, or CFSE^{lo}CD44^{hi}CD62L^{hi} fraction for iT_{CM} cells were sorted by FACS, similar to **Fig.3c**. These cells were then re-labeled with CFSE and transferred into the irradiated recipient mice (2nd transfer). Twenty days later, the same procedure was performed as 3rd transfer. Mice of 3rd transfer were analyzed on day 10 after transfer. The actual numbers of the recovered CFSE^{lo} cells after 1st and 2nd transfer and the number of whole cells

recovered after 3rd transfer are now shown in new **Fig. 3e lower panels**. This protocol is now described in M&M section and briefly in Figure legend.

10. The representation of the result in Figure 5b does not allow me to assess how efficient the conversion into human iTSCM is, because the CD45RA/RO profiles shown have been pre-gated on CCR7+CD95+ cells. What proportion of the cultures is CCR7+CD95+?

We appreciate your comment. We added the gating of CCR7/C95 in new **Fig.5b**. After OP9-hDL1 coculture, approximately 20% of CD4⁺ cells and 85% of CD8⁺ T cells were CCR7⁺C95⁺ cells. Among them, 30-45% was CD45RA⁺CD45RO⁻ for CD4⁺ cells and 14-78% were CD45RA⁺CD45RO⁻ for CD8⁺ cells, depending on the donors. Therefore, conversion efficiency from activated T cells to iT_{SCM} is 5-9% for CD4⁺ T cells and 12-66% for CD8⁺ T cells. This was mentioned in the Result section.

11. What is the fold expansion in Figure 5d?

The fold expansions of in **Figure 5d** are approximately 1-fold for iT_{CM} and 3-folds for iT_{SCM}. We have changed the bar graph in new **Figure 5d** from actual cell number to fold change.

12. A value is missing for the Dltx1 mRNA graph in Supplementary figure 2d. Was this undetectable? What was the lower limit of detection? Are these data from a single experiment?

Supplementary Figure 2d *Dltx1* mRNA of OP9 group was detectable, although it was very low. We have changed the graph style to visualize the OP9 group plots.

Actual value of OP9 group is $0.0009753 \pm 8.863 \times 10^{-5}$, n=3. We performed independent two experiments and detected *Dltx1* mRNA at 12, 24, 36 hours after coculture. *Dltx1* mRNA in OP9 groups was detectable, but very low in all time point.

13. The number of cells recovered in Supplementary Figure 6c is very low (2.5% of input). What is the evidence that the cells expanded?

We agree with the reviewer. The number of cell recovered in this experiment was very low compared with **Fig.3b** for CD4⁺T cells. We repeated the same experiments and found that only small fraction of CD8⁺ T cells were activated by OVA/IFA immunization, probably due

to incomplete cross-presentation of DCs by IFA. Therefore, we removed this data and replaced with experiments using *in vivo* immunization by OVA-expressing tumor cells (new **Supplementary Fig.12a**). In this experiment, we could detect CFSE dilution of iT_{CM} and iT_{SCM} cells (upper panels) and substantial numbers of cells were recovered in the secondary lymphoid organs (lower panels). We confirmed that CD8⁺ iT_{SCM} cells can expand much better than iT_{CM} cells in mice and human.

REVIEWERS' COMMENTS:

Reviewer #1 (Remarks to the Author):

The authors have answered all of my substantial criticisms and I am happy to endorse the manuscript for publication.

Reviewer #2 (Remarks to the Author):

I find that the authors have adequately addressed the issues I raised. I am still a bit puzzled by the poor performance of the Tcm cells (both the in vitro raised as well as the ex vivo population), but the authors have done the experiment I asked for and I cannot argue with the result. I have a few small issues that can easily be dealt with. Nonetheless, I believe that the major claims of the article are well supported by the data and I think there will be quite some interest in this study.

Issues remaining:

1. not all figure legends (especially in the supplementary file) have complete reproducibility statements (numbers of samples, repeats). This is also true for the gene chip results.
2. above the dot plots in figure 5b, it says that the cells were gated on CD8a, but I doubt that this also applies to the CD4 T cells.
3. the legend of supplementary figure 7d does not clearly describe how the experiment shown was performed.
4. are the transcriptome comparisons shown in supplementary figure 16 based on human or mouse cells?

Point-by-point response to Reviewers' Comments:

Reviewer #2

1. not all figure legends (especially in the supplementary file) have complete reproducibility statements (numbers of samples, repeats). This is also true for the gene chip results.

We added the number of samples and reproducibility statements in all figure legends, including microarray data. Sentence about reproducibility in legends were highlighted by yellow color.

2. above the dot plots in figure 5b, it says that the cells were gated on CD8a, but I doubt that this also applies to the CD4 T cells.

Sorry about our mistake. We revised and added the gating label.

3. the legend of supplementary figure 7d does not clearly describe how the experiment shown was performed.

We are sorry for our confusing legend of Supple Fig 7d. We now describe the experimental procedure of Supple Fig 7d in detail in the legend.

4. are the transcriptome comparisons shown in supplementary figure 16 based on human or mouse cells?

The transcriptome comparison was performed using by human samples. We mentioned this in the main text and legends and highlighted by yellow.